# CylinderSplat: 3D Gaussian Splatting with Cylindrical Triplanes for Panoramic Novel View Synthesis

**Qiwei Wang[1]**    **Xianghui Ze[2]**    **Jingyi Yu[1]**    **Yujiao Shi[1]**

[1]Shanghaitech University    [2]Nanjing University of Science and Technology

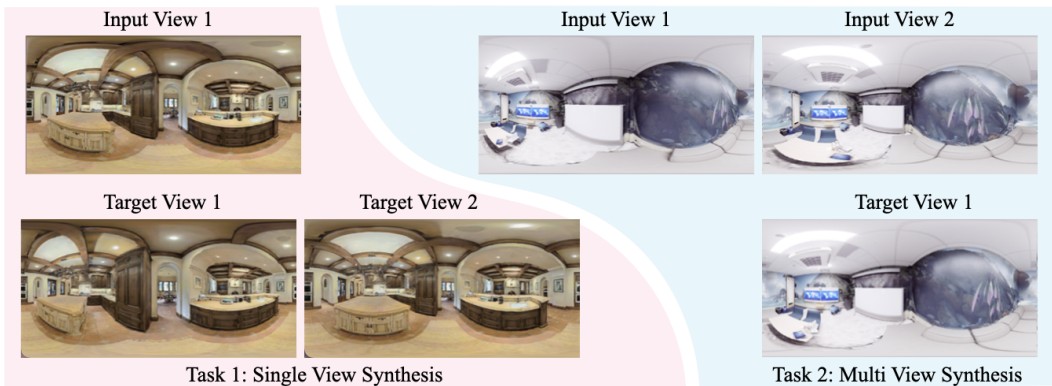

Figure 1: This paper introduces CylinderSplat, a feed-forward panoramic 3D Gaussian Splatting (3DGS) framework for panoramic novel view synthesis from single (left) or sparse (right) input views.

## Abstract

Feed-forward 3D Gaussian Splatting (3DGS) has shown great promise for real-time novel view synthesis, but its application to panoramic imagery remains challenging. Existing methods often rely on multi-view cost volumes for geometric refinement, which struggle to resolve occlusions in sparse-view scenarios. Furthermore, standard volumetric representations like Cartesian Triplanes are poor in capturing the inherent geometry of $360°$ scenes, leading to distortion and aliasing.

In this work, we introduce CylinderSplat, a feed-forward framework for panoramic 3DGS that addresses these limitations. The core of our method is a new cylindrical Triplane representation, which is better aligned with panoramic data and real-world structures adhering to the Manhattan-world assumption. We use a dual-branch architecture: a pixel-based branch reconstructs well-observed regions, while a volume-based branch leverages the cylindrical Triplane to complete occluded or sparsely-viewed areas. Our framework is designed to flexibly handle a variable number of input views, from single to multiple panoramas. Extensive experiments demonstrate that CylinderSplat achieves state-of-the-art results in both single-view and multi-view panoramic novel view synthesis, outperforming previous methods in both reconstruction quality and geometric accuracy. Our code is available at https://github.com/wangqww/CylinderSplat.

## 1 Introduction

The proliferation of 360-degree cameras and the advancement of virtual reality (VR) technologies have spurred significant interest in panoramic imaging. Panoramas offer a complete, immersive field of view, making them an ideal medium for VR applications (Li et al., 2024a; Yu et al., 2025; Luo et al., 2025) and a highly efficient data source for autonomous driving (Shi et al., 2019; 2020; Zhu et al., 2021). A key challenge in this domain is novel view synthesis (NVS), which aims to render photorealistic images from arbitrary viewpoints, providing users with a truly immersive and interactive experience.

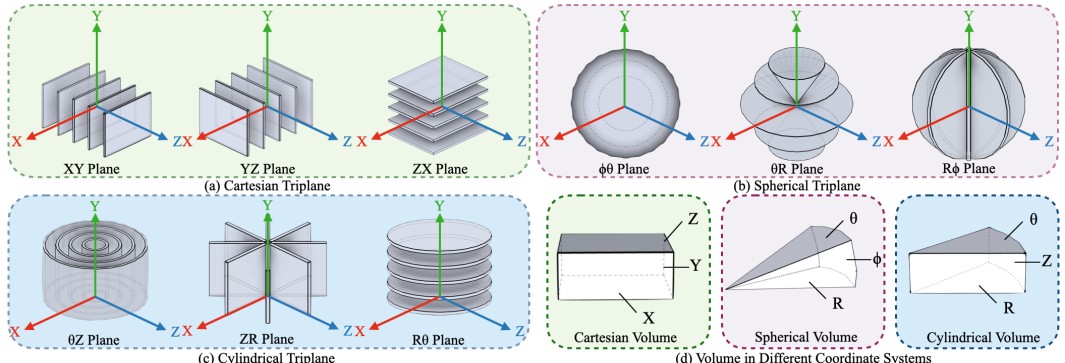

Figure 2: Visualization of the Triplane representation in (a) Cartesian, (b) Spherical, and (c) Cylindrical coordinate systems. (d) The corresponding unit volume elements for each system.

Recently, 3D Gaussian Splatting (3DGS) (Kerbl et al., 2023) has emerged as a breakthrough for real-time, high-fidelity novel view synthesis, with methods falling into two main paradigms. Optimization-based approaches (Kerbl et al., 2023; Lu et al., 2024; Liu et al., 2024; Huang et al., 2024a) achieve exceptional quality via meticulous, per-scene optimization of millions of Gaussian parameters. However, this process is computationally intensive, requiring minutes to hours of training per scene and offering no ability to generalize. In contrast, feed-forward methods (Charatan et al., 2024; Chen et al., 2024; Xu et al., 2025; Wei et al., 2025) leverage pre-trained deep neural networks to predict the Gaussian parameters in a single forward pass, enabling near real-time reconstruction and generalization across scenes.

While 3DGS excels with pinhole cameras, adapting it to the unique geometry of panoramic imagery remains an active area of research. Recent efforts fall into two categories: per-scene optimization frameworks like (Huang et al., 2025a), and generalizable feed-forward methods such as (Zhang et al., 2025; Chen et al., 2025). These feed-forward models typically predict an initial depth prior using a depth foundation model, then refine it based on the cost-volume, but often fail in the presence of occluded regions, resulting in inaccurate geometry and artifacts. Although alternative volumetric representations like Triplanes have been explored for refinement (e.g. (Wei et al., 2025)), these solutions are tailored for pinhole cameras.

In this work, we introduce **CylinderSplat**, a new feed-forward framework for panoramic 3DGS that resolves the aforementioned limitations, as shown in Fig. 1, through a dual-branch architecture.

Our first branch, the pixel branch, is inspired by recent advances in 3D reconstruction (Wang et al., 2024; 2025a;b; Yang et al., 2025a). It employs self-attention within frames and cross-attention among frames to aggregate multi-view information and produce a feature point cloud. This design allows our network to flexibly handle an arbitrary number of input views and predict Gaussian parameters corresponding to each input pixel. However, while the pixel branch generates high-quality Gaussians for well-observed regions, it fails in sparse-view scenarios where large baselines leave occluded areas without point cloud coverage. This results in significant holes and non-uniformity in the reconstructed scene, necessitating a dedicated mechanism for geometric completion.

To address this limitation, our second branch, the volume branch, which complements the pixel branch, introduces a new cylindrical Triplane representation at each camera's location. This Triplane defines a local volume, centered at the camera, that represents a dense grid of points uniformly distributed in a $360°$ cylindrical space. Its purpose is to correct geometric errors and hallucinate plausible details within the occluded regions of the pixel branch's output. To maintain efficiency, we use the Triplane to compress this dense 3D feature grid. Instead of storing features for all $\Theta \times Z \times R$ points, the Triplane reduces the storage complexity from $O(\Theta \cdot Z \cdot R)$ to $O(\Theta \cdot Z + Z \cdot R + R \cdot \Theta)$. This representation is initialized with features from the corresponding view's pixel branch and is further enhanced through triplane attention mechanisms.

The choice of a cylindrical coordinate system for our Triplane is at the core of our approach. Our inspiration comes from physics, where the choice of orthogonal curvilinear coordinates (such as spherical and cylindrical) (Fig. 2(b) and (c)) simplifies complex problems with symmetries. Our task is similar: we optimize 3D Gaussians distributed in a $360°$ space around a central camera, so the spherical and cylindrical coordinate systems are a natural comparison to the Cartesian sys-

tems(Fig. 2(a)). On the other hand, most of the scenes in the real world adhere to the Manhattan-world assumption (Coughlan & Yuille, 1999)—that orthogonal surfaces dominate urban and indoor environments. A spherical Triplane (Fig. 2(b)) struggles to model these simple planes, but a cylindrical Triplane (Fig. 2(c)) is exceptionally well-suited, as its $ZR$ and $R\Theta$ planes naturally align with the vertical walls and horizontal floors prevalent in man-made environments.

The synergy between the flexible pixel branch and the geometrically aware volume branch, which constructs a local cylindrical Triplane for each input camera, enables CylinderSplat to robustly handle a varying number of inputs, ranging from single to multiple. We provide extensive analysis in our experiments to validate the superiority of the cylindrical Triplane and demonstrate that our overall framework outperforms previous methods. Our primary contributions are summarized as follows:

- A new cylindrical Triplane representation, compliant with the Manhattan-world assumption, designed to capture the unique geometric properties of panoramic images.
- A dual-branch feed-forward framework, CylinderSplat, for panoramic 3DGS, combines pixel-based reconstruction for observed regions with volume-based completion for occluded areas, enabling robust novel view synthesis from single or multiple inputs.
- State-of-the-art performance in both quality and geometry accuracy for single- and multi-view panoramic novel view synthesis.

## 2    RELATED WORK

**Novel View Synthesis for Pinhole Cameras.**    Novel view synthesis has rapidly progressed from NeRF methods (Mildenhall et al., 2021) to real-time, high-fidelity 3D Gaussian Splatting (3DGS) (Kerbl et al., 2023). 3DGS methods fall into two main paradigms: computationally expensive **per-scene optimization** techniques that achieve exceptional quality (Kerbl et al., 2023; Lu et al., 2024; Lin et al., 2024; Liu et al., 2024; Wu et al., 2025), and generalizable **feed-forward** approaches that use pre-trained networks (Charatan et al., 2024; Chen et al., 2024; Wei et al., 2025). While recent feed-forward methods have scaled to large scenes (Huang & Mikolajczyk, 2025; Cheng et al., 2025; Wang et al., 2025c; Jiang et al., 2025), all these 3DGS techniques are designed for pinhole cameras and fail to handle the unique geometric distortions of panoramic imagery. Our work addresses this gap with a framework specifically engineered for panoramic data.

**Novel View Synthesis for Panoramas.**    Novel view synthesis for 360° panoramas, a challenging task crucial for immersive VR, has shifted from slow NeRF-based methods (Chen et al., 2023) to 3DGS. Current optimization-based 3DGS approaches (Yang et al., 2025b; Li et al., 2025; Huang et al., 2025a;b; Zhou et al., 2024; Pu et al., 2024) achieve high-quality direct panoramic rendering, but are constrained by the cost of per-scene optimization. Conversely, faster feed-forward methods (Zhang et al., 2025; Chen et al., 2025) are often limited to two input views, struggle with occlusions, and rely on inefficient indirect rendering pipelines. Our work bridges this gap by introducing a feed-forward framework that adopts a direct panoramic representation, utilizing a new Triplane-based module to handle occlusions and refine geometry explicitly.

**Triplane Representations in Novel View Synthesis.** Triplane-based representations have become a popular technique for encoding 3D scenes (Chan et al., 2022; Shue et al., 2023; Zou et al., 2024; Ju & Li, 2025; Xu et al., 2024; Zhan et al., 2025). However, their application has been limited. Many approaches are confined to small-scale objects (Chan et al., 2022; Zou et al., 2024; Ju & Li, 2025) or rely on computationally expensive diffusion priors (Shue et al., 2023; Ju & Li, 2025). While some methods can handle large scenes (Xu et al., 2024; Zhan et al., 2025), they are restricted to per-scene optimization. A notable exception is OmniScene (Wei et al., 2025), a feed-forward method for large-scale pinhole reconstruction. Despite its innovation, OmniScene's Cartesian Triplane is ill-suited for panoramic geometry, does not support multi-frame inputs, and can produce overly smooth renderings. To address this issue of blurriness, we introduce an RGB retrieval strategy that directly queries high-frequency features from the input views, enhancing our Triplane-based renderings.

## 3    METHOD

We propose a dual-branch (pixel, volume) feed-forward architecture for 3DGS reconstruction, optimized with a three-stage curriculum. First, a pixel branch (Sec. 3.1) is trained to establish a high-

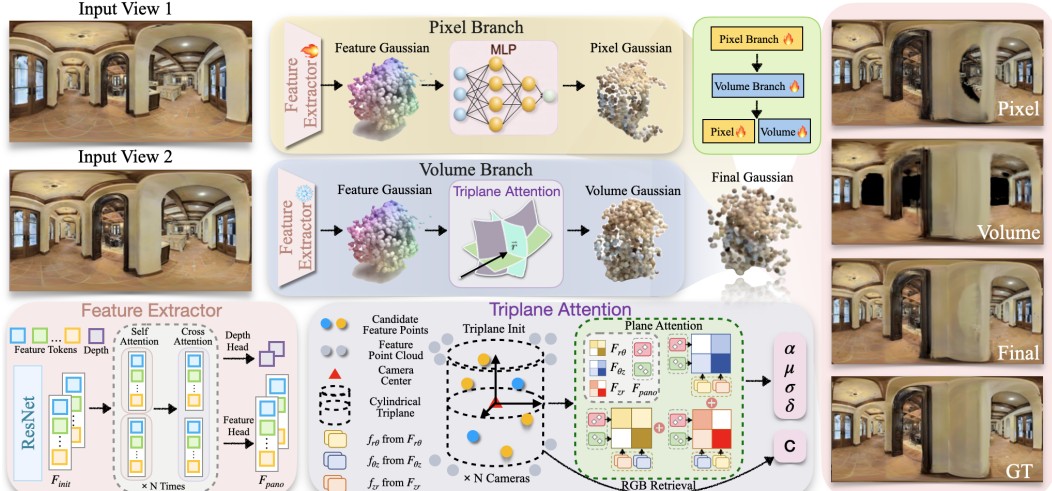

Figure 3: **Overview of our CylinderSplat framework**. Our method uses a dual-branch architecture trained via a three-stage curriculum. The pixel branch uses a multi-view attention mechanism to generate high-quality Gaussians for well-observed regions. The volume branch is designed to fill the gaps by lifting features into our cylindrical triplane representation, thereby completing the scene geometry robustly. The outputs from both branches are then unified for a final render.

quality baseline for well-observed regions. Next, with the pixel branch frozen, a volume branch (Sec. 3.2) using a new cylindrical Triplane is trained to provide robust geometric completion for sparse and occluded areas. Finally, both branches are jointly fine-tuned to merge the pixel branch's detail with the volume branch's completeness into a single high-fidelity scene. The entire process is supervised by a composite loss function (Sec. 3.3), and our choice of the cylindrical Triplane is justified in the supplementary materials C.

## 3.1 PIXEL BRANCH

Prior feed-forward methods (Zhang et al., 2025; Chen et al., 2025) typically follow a pipeline where they first use a depth foundation model to predict a geometry prior, then refine this prior with a multi-view cost volume, and finally generate a feature point cloud ($P_{\text{feat}}$) from which to predict Gaussian parameters. However, this reliance on a cost-volume is computationally expensive and architecturally inflexible, as it requires a fixed number of input views and necessitates retraining to change the number of input views.

Our pixel branch is designed to overcome these specific limitations. While we also leverage an initial depth prior from UniK3D (Piccinelli et al., 2025), we replace the cost volume with a more efficient attention-based mechanism, corresponding to our framework(Fig 3)'s Feature Extractor, inspired by (Wang et al., 2025a). We first use a ResNet and a stack of $L = 6$ attention layers to aggregate a rich, multi-view context. This network then predicts a refined depth map ($D_{\text{pano}}$), alongside a feature map ($F_{\text{pano}}$). Using the refined depth, we unproject each pixel to its 3D coordinate to form $P_{\text{feat}}$, where each point is endowed with the corresponding feature from $F_{\text{pano}}$. Finally, these features are decoded into the complete set of Gaussian parameters, $\mathcal{G}_{\text{pixel}}$. This camera-agnostic process is memory-efficient and provides a strong baseline for the volume branch to refine further.

## 3.2 VOLUME BRANCH

The volume branch is designed to complete the geometry in under-observed and occluded regions by employing an efficient Triplane representation (Zou et al., 2024; Wei et al., 2025), which reduces memory complexity from $O(N^3)$ to $O(N^2)$. At the core of our approach is the replacement of the standard axis-aligned Cartesian planes used in prior methods with a new **cylindrical Triplane** that is better suited to the geometry of $360°$ panoramic scenes. For each input view's camera position, we initialize an independent, local Triplane within a cylindrical volume bounded by dimensions ($R_0$, $\Theta_0$, $Z_0$), as shown in Fig. 3(a). Each Triplane's three orthogonal feature planes ($F_{r\theta}$, $F_{\theta z}$, and $F_{zr}$) are first initialized with a set of learnable grid embeddings. Subsequently, as illustrated in Fig. 3(b),

we populate these planes by aggregating the features of any points from the pixel branch that fall within the Triplane's volume. These view-specific Triplanes are then processed in parallel, each undergoing refinement via Cross-Plane and Triplane-to-Image Attention (Wei et al., 2025; Huang et al., 2023; Li et al., 2024b) to decode a set of Gaussian parameters. Finally, the Gaussians generated from all individual Triplanes are concatenated to form the complete output of the Volume Branch.

### 3.2.1 Cross-Plane Attention

Following initialization, the three feature planes($F_{r\theta}$, $F_{\theta z}$, and $F_{zr}$) from the same cylinder must exchange information to form a cohesive 3D representation. We achieve this through our cross-plane attention mechanism, where each feature on one plane queries corresponding features from the other two planes along the orthogonal dimension. For example, consider updating a feature $\mathbf{f}_{\theta z}(i, j)$ in the $F_{\theta z}$ plane, we sample $N_r$ points along the corresponding radial axis, indexed by $k \in \{0, \ldots, N_r - 1\}$, and retrieve features from the other two planes($F_{r\theta}$ and $F_{zr}$) at these locations to serve as the keys and values. An attention mechanism then computes a weighted sum of these values based on their similarity to the query. This update process, including a residual connection, is executed in parallel for all features and can be conceptually expressed as:

$$\mathbf{f}'_{\theta z}(i, j) = \mathbf{f}_{\theta z}(i, j) + \sum_{k=0}^{N_r - 1} \left( w_{zr}^{(ijk)} \mathbf{f}_{zr}(j, k) + w_{r\theta}^{(ijk)} \mathbf{f}_{r\theta}(k, i) \right). \tag{1}$$

Here, $\mathbf{f}'_{\theta z}(i, j)$ is the updated feature, and the coefficients $w$ are the aggregation weights produced by the attention's softmax over the query-key scores. This comprehensive fusion ensures a unified spatial representation.

### 3.2.2 Triplane-to-Image Attention

To enrich the Triplane features with visual evidence from source images, we perform an additional refinement step using Triplane-to-Image Attention. In this mechanism, the triplane features fused in the attention step of the cross-plane attention act as queries, while the panoramic image features $F_{pano}$ from the pixel branch serve as the keys and values. For example, in the update process for $\mathbf{f}'_{\theta z}(i, j)$ in the $F'_{\theta z}$ plane, we sample $N'_r$ points along the orthogonal radial dimension, indexed by $k \in \{0, \ldots, N'_r - 1\}$, and project each resulting 3D point $(\theta_i, z_j, r'_k)$ into the panorama to obtain its corresponding pixel coordinates $(u_{ijk}, v_{ijk})$. The characteristic of the panoramic image $\mathbf{f}_{pano}^{(ijk)}$ at this location is then recovered from $F_{pano}$. This set of features is aggregated into a single context vector via the cross-attention mechanism and added back to the original query through a residual connection. This update rule can be conceptually summarized as follows:

$$\mathbf{f}''_{\theta z}(i, j) = \mathbf{f}'_{\theta z}(i, j) + \sum_{k=0}^{N'_r - 1} w_{pano}^{(ijk)} \mathbf{f}_{pano}^{(ijk)}. \tag{2}$$

Here, $\mathbf{f}''_{\theta z}(i, j)$ is the final visually enriched feature, and the learned attention weights $w_{pano}$ ensure that the final 3D representation is closely aligned with the 2D input images.

### 3.2.3 Decoding Gaussians from the Triplane

For each per-camera Triplane, we generate the corresponding volume Gaussian primitives by first uniformly sampling a dense grid of $N_r \times N_\theta \times N_z$ points within its cylindrical volume. At each grid point $(\theta_i, z_j, r_k)$, we compute its feature $\mathbf{f}_{r\theta z}$ by querying and summing the corresponding features from the three refined tri-planes: $\mathbf{f}''_{r\theta}(k, i) + \mathbf{f}''_{\theta z}(i, j) + \mathbf{f}''_{zr}(j, k)$. A MLP (Rosenblatt, 1958) then processes this aggregated feature to predict a set of *local*, normalized parameters for each potential Gaussian: $\{\boldsymbol{\delta}_{local}, \mathbf{S}_{local}, \mathbf{R}, \alpha\}_{volume} = \text{MLP}(\mathbf{f}_{r\theta z})$. Here, $\boldsymbol{\delta}_{local} = (\delta_r, \delta_\theta, \delta_z)$ represents positional offsets and $\mathbf{S}_{local} = (S_r, S_\theta, S_z)$ represents anisotropic scaling factors, both defined within the local cylindrical coordinate frame of the grid cell.

Since the projection coordinate system for 3DGS remains Cartesian, we still need to transform these locally defined cylindrical attributes into Cartesian coordinates. The final position $\mathbf{x}' = (x', y', z')$ is calculated by applying the learned offsets $\boldsymbol{\delta}_{local}$ to the base cylindrical coordinates of the grid

cell $(r, \theta, z)$ and then performing a standard cylindrical-to-Cartesian conversion:

$$(x', y', z') = (-(r + \delta_r)\sin(\theta + \delta_\theta), \quad z + \delta_z, \quad -(r + \delta_r)\cos(\theta + \delta_\theta)). \tag{3}$$

Crucially, the local scaling factors $\mathbf{S}_{\text{local}}$ must also be transformed to represent the Gaussian shape in Cartesian space correctly. This is achieved using the Jacobian matrix $\mathbf{J}$ of the coordinate transformation. The final anisotropic scale $\mathbf{S}'$ is computed as:

$$\mathbf{S}' = |\mathbf{J}| \cdot \mathbf{S}_{\text{local}}, \tag{4}$$

where $|\mathbf{J}|$ is the matrix of the absolute values of the Jacobian's elements, and $\mathbf{J}$ is defined as:

$$\mathbf{J} = \begin{pmatrix} -\sin(\theta + \delta_\theta) & -(r + \delta_r)\cos(\theta + \delta_\theta) & 0 \\ 0 & 0 & 1 \\ -\cos(\theta + \delta_\theta) & (r + \delta_r)\sin(\theta + \delta_\theta) & 0 \end{pmatrix}. \tag{5}$$

### 3.2.4 VOLUME RGB RETRIEVAL

Since the features derived from the Triplane are high-level and semantic, they often lack the high-frequency details necessary for photorealistic color. To address this, we determine the color $C$ for each volume Gaussian via an RGB Retrieval mechanism inspired by (Miao et al., 2025), which leverages information directly from the source images.

For each Gaussian, we project its center $\mathbf{x}'$ into all $N_v$ source views to retrieve the pixel colors $\{C_v\}$. To handle occlusions, we compute a visibility score for each view, $s_v = d_g - d_o$, where $d_g$ is the distance of the Gaussian from its corresponding camera and $d_o$ is the reference depth from Unik3D at that pixel. A smaller score indicates higher visibility. A final MLP then predicts the definitive color based on a visibility-weighted aggregation of these retrieved colors:

$$C = \text{MLP}\left(\sum_{v=1}^{N_v} w_v \cdot C_v\right), \quad \text{where } w_v = \text{softmax}(-s_v). \tag{6}$$

Here, $C$ is the final predicted color and $w_v$ is the learned visibility weight. This visibility-aware approach encourages the model to learn color from the most reliable, unoccluded views, providing an implicit supervisory signal to align the Gaussian geometry with the depth priors. The full parameter set for each Triplane-based Gaussian is thus given by $\{\mathbf{x}', \mathbf{S}', \mathbf{R}, C, \alpha\}_{\text{volume}}$. Finally, we concatenate the Gaussians from all per-camera Triplanes to form the complete output of our volume branch, $\mathcal{G}_{\text{volume}}$.

## 3.3 LOSS FUNCTION

Our model is optimized using a composite rendering loss, $\mathcal{L}_{\text{render}}$, applied consistently across our three-stage training curriculum. This loss enforces photometric accuracy, perceptual realism, and geometric consistency. It is a weighted sum of three components:

$$\mathcal{L}_{\text{render}} = \left\| \hat{I} - I_{\text{gt}} \right\|_1 + 0.05 * \mathcal{L}_{\text{LPIPS}}(\hat{I}, I_{\text{gt}}) + 0.1 * \left\| \hat{D} - D_{\text{ref}} \right\|_1, \tag{7}$$

where $\hat{I}$ and $\hat{D}$ are the rendered image and depth map, $I_{\text{gt}}$ is the ground-truth image, and $D_{\text{ref}}$ is the reference depth map from UniK3D. The source of the rendered outputs changes with each training stage: first using only pixel branch Gaussians ($\mathcal{G}_{\text{pixel}}$), then only volume branch Gaussians ($\mathcal{G}_{\text{volume}}$), and finally the union of both ($\mathcal{G}_{\text{pixel}} \cup \mathcal{G}_{\text{volume}}$) for joint fine-tuning.

## 4 EXPERIMENTS

**Datasets**. To ensure a fair comparison with prior feed-forward panoramic methods, we follow the experimental setup of (Zhang et al., 2025; Chen et al., 2023). We evaluate our model on three synthetic datasets—Matterport3D (Chang et al., 2017), Replica (Straub et al., 2019), and Residential (Habtegebrial et al., 2022)—and one real-world dataset, 360Loc (Huang et al., 2024b). All experiments are conducted using images with a resolution of $512 \times 1024$.

Table 1: Quantitative comparison for the two-view reconstruction on the Matterport3D, Replica, and Residential datasets. The first, second, and third best results are highlighted. Methods marked with * indicate that we reimplemented them using their official code.

| Dataset | Matterport3D | | | | | | | | | | | | Replica | | | | Residential | | | |
|---|---|---|---|---|---|---|---|---|---|---|---|---|---|---|---|---|---|---|---|---|
| Baseline | 2.0m | | | | 1.5m | | | | 1.0m | | | | 1.0m | | | | about 0.3m | | | |
| Method | PCC↑ | WS-PSNR↑ | SSIM↑ | LPIPS↓ | PCC↑ | WS-PSNR↑ | SSIM↑ | LPIPS↓ | PCC↑ | WS-PSNR↑ | SSIM↑ | LPIPS↓ | PCC↑ | WS-PSNR↑ | SSIM↑ | LPIPS↓ | PCC↑ | WS-PSNR↑ | SSIM↑ | LPIPS↓ |
| MVSplat | — | 13.31 | 0.595 | 0.554 | — | 21.82 | 0.807 | 0.230 | — | 28.19 | 0.912 | 0.105 | — | 30.54 | 0.958 | 0.059 | — | 31.21 | 0.906 | 0.200 |
| PanoGRF | — | 20.96 | 0.701 | 0.352 | — | 23.38 | 0.811 | 0.282 | — | 27.12 | 0.876 | 0.195 | — | 29.22 | 0.937 | 0.134 | — | 31.03 | 0.909 | 0.207 |
| OmniScene* | 0.732 | 22.75 | 0.707 | 0.241 | 0.788 | 23.73 | 0.749 | 0.204 | 0.846 | 25.43 | 0.797 | 0.151 | 0.811 | 27.14 | 0.872 | 0.177 | 0.762 | 27.61 | 0.857 | 0.195 |
| Splatter360* | 0.684 | 21.31 | 0.741 | 0.285 | 0.791 | 24.68 | 0.801 | 0.199 | 0.887 | 26.71 | 0.828 | 0.141 | 0.643 | 25.52 | 0.879 | 0.134 | 0.712 | 28.15 | 0.860 | 0.261 |
| PanSplat | 0.716 | 20.56 | 0.777 | 0.265 | 0.779 | 24.09 | 0.849 | 0.181 | 0.819 | 28.83 | 0.935 | 0.091 | 0.681 | 30.78 | 0.962 | 0.069 | 0.744 | 30.97 | 0.917 | 0.172 |
| Ours | 0.851 | 23.76 | 0.835 | 0.175 | 0.867 | 25.91 | 0.873 | 0.128 | 0.923 | 28.89 | 0.937 | 0.081 | 0.885 | 30.29 | 0.959 | 0.057 | 0.828 | 28.17 | 0.866 | 0.156 |

Table 2: Quantitative comparison for the single-view reconstruction task.

| Dataset | Matterport3D | | | | | | | | | | | | Replica | | | | Residential | | | |
|---|---|---|---|---|---|---|---|---|---|---|---|---|---|---|---|---|---|---|---|---|
| Baseline | 2.0m | | | | 1.5m | | | | 1.0m | | | | 1.0m | | | | about 0.3m | | | |
| Method | PCC↑ | WS-PSNR↑ | SSIM↑ | LPIPS↓ | PCC↑ | WS-PSNR↑ | SSIM↑ | LPIPS↓ | PCC↑ | WS-PSNR↑ | SSIM↑ | LPIPS↓ | PCC↑ | WS-PSNR↑ | SSIM↑ | LPIPS↓ | PCC↑ | WS-PSNR↑ | SSIM↑ | LPIPS↓ |
| OmniScene* | 0.681 | 22.52 | 0.707 | 0.244 | 0.757 | 23.50 | 0.749 | 0.206 | 0.816 | 25.22 | 0.798 | 0.152 | 0.775 | 24.91 | 0.861 | 0.137 | 0.754 | 26.97 | 0.820 | 0.177 |
| Splatter360* | 0.628 | 21.81 | 0.650 | 0.380 | 0.648 | 22.40 | 0.690 | 0.350 | 0.653 | 24.15 | 0.738 | 0.308 | 0.631 | 24.36 | 0.805 | 0.244 | 0.528 | 29.73 | 0.805 | 0.261 |
| PanSplat* | 0.596 | 19.72 | 0.625 | 0.334 | 0.578 | 20.84 | 0.668 | 0.292 | 0.621 | 22.43 | 0.717 | 0.223 | 0.601 | 23.32 | 0.803 | 0.178 | 0.485 | 26.59 | 0.814 | 0.193 |
| Ours | 0.821 | 23.75 | 0.822 | 0.175 | 0.856 | 25.13 | 0.854 | 0.136 | 0.903 | 27.01 | 0.915 | 0.089 | 0.864 | 26.14 | 0.887 | 0.103 | 0.805 | 27.87 | 0.843 | 0.154 |

Table 3: Quantitative comparison for the two-view reconstruction task on the 360Loc dataset.

| Dataset | 360Loc (avg. 1.40m baseline) | | | |
|---|---|---|---|---|
| Method | PCC↑ | WS-PSNR↑ | SSIM↑ | LPIPS↓ |
| MVSplat | — | 24.67 | 0.823 | 0.170 |
| OmniScene* | 0.830 | 27.41 | 0.831 | 0.153 |
| Splatter360* | 0.866 | 28.14 | 0.860 | 0.127 |
| PanSplat | 0.188 | 28.24 | 0.873 | 0.108 |
| Ours | 0.884 | 28.35 | 0.896 | 0.095 |

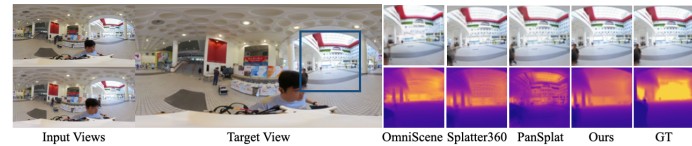

Figure 4: Qualitative comparison of 360Loc (two-view). Left: input/target views. Right: zoomed-in novel views and depth maps (warm = far, cool = near). Our ground truth (GT) depth is obtained from DepthAnywhere (Wang & Liu, 2024), which serves as a reference for calculating PCC.

For the **synthetic datasets**, each represented by a three-frame sequence, we train our model on 20,000 sequences from Matterport3D, where adjacent frames are separated by 0.5m. We then test on held-out sequences from Matterport3D, Replica, and Residential with varying baselines from 0.3m to 2.0m. For the two-view reconstruction task, consistent with prior work (Zhang et al., 2025; Chen et al., 2023), we use the first and last frames of each sequence as input (1.0m baseline) to reconstruct the middle frame as the target. To validate our model's flexibility, we also conduct a single-view reconstruction experiment by fine-tuning our two-view model. For this task, we use the middle frame of each sequence as the sole input to predict the two outer frames as rendering targets.

For the **real-world dataset**, 360Loc (Huang et al., 2024b) includes four scenes captured in diverse campus-scale indoor and outdoor environments, comprising 18 sequences of continuous frames captured at 0.46m intervals. We follow (Zhang et al., 2025)'s fine-tuning and evaluation setting. We first fine-tune our pre-trained two-view model (trained on the synthetic datasets) on three scenes (13 sequences) from 360Loc. Subsequently, we test it on the held-out scene (5 sequences). During both fine-tuning and testing, we use two input frames with a 1.4m separation (skipping two intermediate frames) and render all four views (the two inputs and the two intermediate frames) as output targets.

**Evaluation Metrics**. Consistent with prior work (Chen et al., 2023; Zhang et al., 2025), for **image quality**, we report SSIM (Hore & Ziou, 2010), LPIPS (Zhang et al., 2018), and WS-PSNR (Sun et al., 2017), using WS-PSNR for its robustness to panoramic distortions. For **geometry**, we compute the PCC (Benesty et al., 2009) against reference depths generated by DepthAnywhere (Wang & Liu, 2024), using this scale-invariant metric to provide a measure in the absence of GT depth.

## 4.1 COMPARISONS WITH THE STATE-OF-THE-ART

We conduct a comparison against several state-of-the-art(SOTA) feed-forward methods, following the two-view reconstruction setup from Zhang et al. (2025) for both synthetic and real-world scenes. Our baselines include two panoramic 3DGS models, PanSplat (Zhang et al., 2025) and Splatter360 (Chen et al., 2025), a panoramic NeRF model, PanoGRF (Chen et al., 2023), and two methods adapted from the pinhole domain: the surrounding-view OmniScene (Wei et al., 2025) and the multi-view MVSplat (Chen et al., 2024), as shown in Table 1, Figure 5, Figure 6 and Figure 8. Results for MVSplat, PanoGRF, and PanSplat are taken from Zhang et al. (2025), while OmniScene and Splatter360 were reimplemented using their official code. Since OmniScene only supports pinhole cameras, we adapt them by decomposing each panoramic image via cubemap projection. Additionally, we conduct a single-view reconstruction comparison on the synthetic datasets

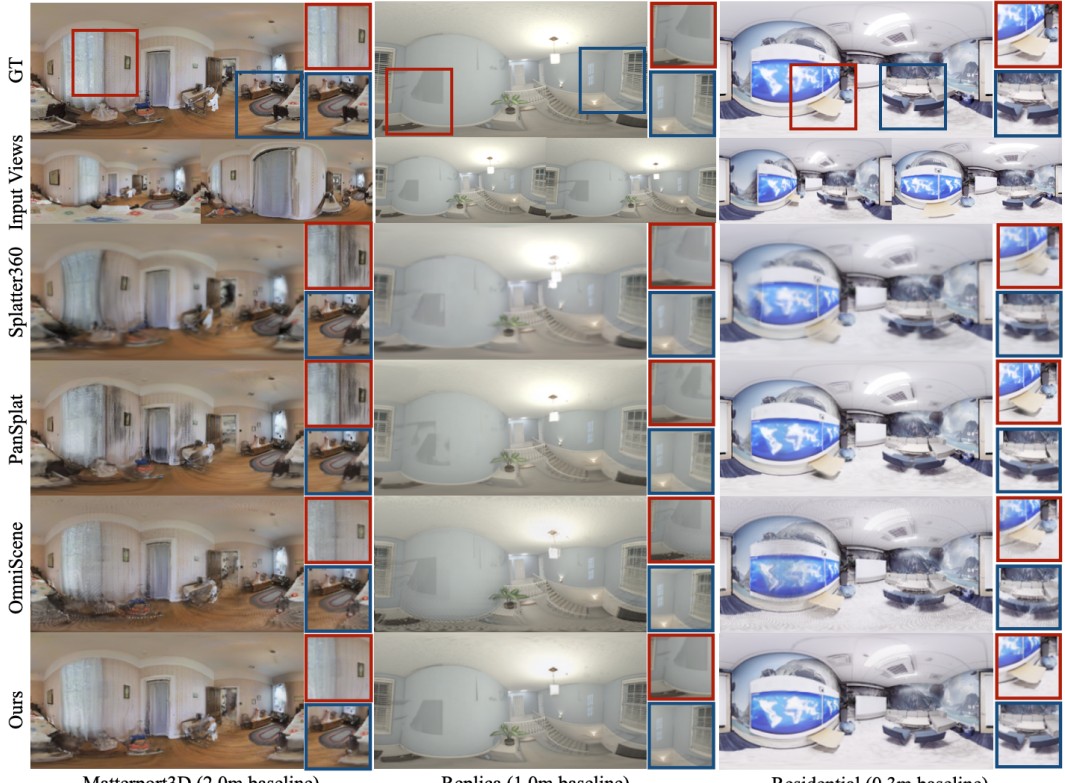

Figure 5: **Qualitative comparison of synthetic scenes for the two-view input task across different baselines.** The top two rows display the target ground truth and input views, followed by comparisons of different methods. Zoomed-in regions highlight the superior completeness, sharpness, and reduced artifacts of our method.

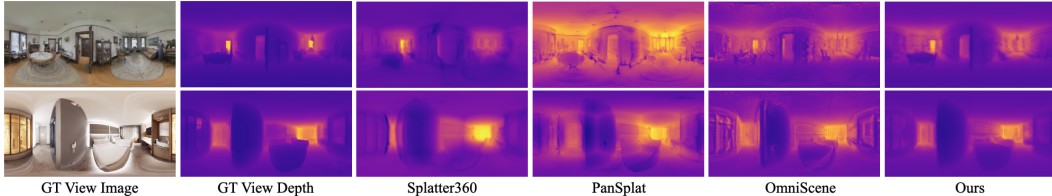

Figure 6: **Qualitative depth comparison on synthetic scenes for the two-view input task.** From left to right: ground-truth RGB, the reference depth from DepthAnywhere (Wang & Liu, 2024), and the depth maps from different methods. Our result shows the best consistency with the reference, particularly on the floor and ceiling.

in Table 2. As the cost-volume mechanisms in PanSplat and Splatter360 require at least two views, we create a fair baseline by fine-tuning them on a duplicated single-view input before comparing them against our approach.

**Analysis.** Compared to cost-volume-based methods like Splatter360 and PanSplat, our approach demonstrates superior geometric accuracy and robustness in challenging scenarios. As shown in our qualitative results (Fig. 4, Fig. 5, Fig. 6 and Fig 8), our Triplane-based completion effectively handles large baselines and distorted regions (e.g., ceilings and floors) where cost-volume methods produce holes, artifacts, and inconsistent depth. Quantitatively (Tables 1 and 2), this advantage is apparent in two-view and especially single-view tasks, where cost-volume methods fail due to their architectural limitations. Our method also outperforms Triplane-based methods, such as OmniScene. OmniScene utilizes a single, central Cartesian Triplane and processes views independently, which introduces distortion artifacts in panoramic renderings (Fig. 4, Fig. 5, Fig. 6 and Fig 8) and limits its multi-view fusion capabilities. In contrast, our framework's use of multiple, per-camera cylindrical Triplanes,

Table 4: **Ablation Study** on Matterport3D (2.0m baseline) using a two-view input. "Pixel Branch" and "Volume Branch" denote using only the respective branch. Rows 2-4 compare the performance of cylindrical, spherical, and Cartesian Triplanes. Row 5 presents results without our RGB retrieval, and row 6 shows the impact of the mutil Triplane strategy. Row 7 shows the result of training without our curriculum.

| Matterport3D (2.0m) | | | | |
|---|---|---|---|---|
| Method | PCC | WS-PSNR | SSIM | LPIPS |
| Only Pixel Branch | 0.813 | 23.21 | 0.817 | 0.179 |
| Only Cylindrical Volume Branch | 0.782 | 22.17 | 0.782 | 0.210 |
| Only Spherical Volume Branch | 0.581 | 19.22 | 0.633 | 0.398 |
| Only Cartesian Volume Branch | 0.564 | 16.77 | 0.495 | 0.545 |
| Only Cylindrical Volume Branch (w/o RGB) | 0.703 | 20.16 | 0.661 | 0.409 |
| Full (w/o Multi Triplane) | 0.826 | 23.47 | 0.805 | 0.194 |
| Full (end to end) | 0.809 | 23.25 | 0.791 | 0.185 |
| Full | **0.851** | **23.76** | **0.835** | **0.175** |

Table 5: **Multi-view results on 360Loc**, extending beyond the initial two-view (1.4m baseline) setup to include 3-view and 4-view configurations.

| 360Loc (avg. 1.40m baseline) | | | | | |
|---|---|---|---|---|---|
| Views | PCC | WS-PSNR | SSIM | LPIPS | GPU |
| 2 | 0.8839 | 28.35 | 0.896 | 0.095 | 7GB |
| 3 | 0.8937 | 28.51 | 0.907 | 0.093 | 13GB |
| 4 | 0.8961 | 28.97 | 0.912 | 0.094 | 20GB |

Table 6: Comparison of Model Complexity and Efficiency. "Inference Time" measures the end-to-end latency for a single forward pass that outputs Gaussians and renders them into one panorama.

| Method | Ours | PanSplat | Splatter360 | OmniScene |
|---|---|---|---|---|
| Parameters | 13.6M | 20.5M | 38.7M | 76.9M |
| Inference Time | 0.29s | 0.32s | 0.54s | 0.48s |

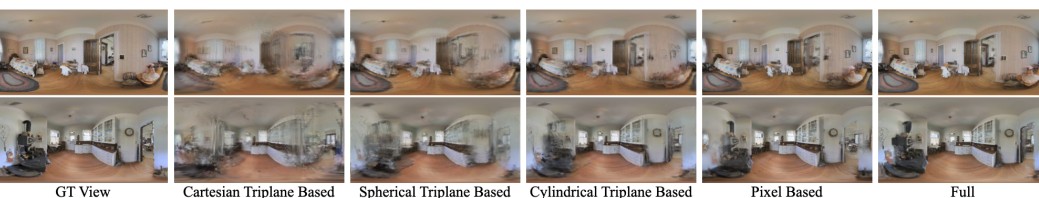

| GT View | Cartesian Triplane Based | Spherical Triplane Based | Cylindrical Triplane Based | Pixel Based | Full |

Figure 7: Ablation study visualizations on Matterport3D (leftmost column is ground truth). Among the different coordinate systems for the Triplane, the Cartesian version shows significant distortion, and the spherical version struggles with distant rooms. Our cylindrical Triplane performs best. While the pixel branch is high-quality in visible areas, combining it with the Triplane branch improves reconstruction of distant regions.

combined with an RGB retrieval strategy and a dedicated fusion mechanism in the pixel branch, yields improved rendering quality and geometric fidelity, particularly in the two-view setting.

## 4.2 ABLATION STUDIES

**Effectiveness of Key Components.** As shown in Table 4 and Fig. 7, our ablation study confirms our key design choices. Our staged training curriculum outperforms using either the pixel and volume branch alone, as well as an end-to-end training approach. The end-to-end method, which emphasizes the pixel branch due to its stronger performance in non-occluded areas, often results in an undertrained volume branch and overall subpar results. When examining the volume branch, the cylindrical Triplane consistently outperforms its Cartesian and spherical counterparts, confirming it as the more effective geometric representation for panoramic scenes. Lastly, we also show that our RGB retrieval mechanism and multi-Triplane strategy outperform OmniScene's original Triplane method (color decoded directly from volume features and only one Triplane at the center of all input views), mainly because our approach better preserves high-frequency image details and effectively manages multi-frame inputs.

**Multi-View Input.** As shown in Table 5, to demonstrate scalability, we fine-tune our two-view model for three- and four-view tasks. The results show that the quality of novel view synthesis improves as the number of input views increases, confirming the effectiveness of our framework in a multi-view setting. We don't compare methods like OmniScene or PanSplat, as their architectures are locked to a fixed number of input cameras and can't be fine-tuned without retraining from scratch.

**Model Efficiency.** As shown in Table 6, by replacing the expensive cost volume with an efficient Triplane representation and avoiding heavy feature extractors such as DINO (Oquab et al., 2023) or U-Net (Ronneberger et al., 2015), our lightweight design is more efficient than competing methods.

## 4.3 VALIDATION AGAINST GROUND TRUTH DEPTH

We primarily used PCC for geometric evaluation (Tables 1, 2, and 3) as dense GT depth is unavailable for the Replica, Residential, and 360Loc test splits. To eliminate metric bias, we evaluated against GT depth on Matterport3D, as shown in Table 7. The results confirm: (1) Consistency: CylinderSplat consistently outperforms baselines across standard metrics (AbsRel, RMSE, $\delta_1$), validating our PCC conclusions. (2) Superiority without Supervision: Unlike competitors (e.g., Splat-

Table 7: Quantitative evaluation of geometric accuracy against Ground Truth (GT) depth on the Matterport3D dataset.

| Method | Matterport3D (2.0m) | | | Matterport3D (1.5m) | | | Matterport3D (1.0m) | | |
|---|---|---|---|---|---|---|---|---|---|
| | AbsRel ↓ | RMSE ↓ | $\delta_1$ ↑ | AbsRel ↓ | RMSE ↓ | $\delta_1$ ↑ | AbsRel ↓ | RMSE ↓ | $\delta_1$ ↑ |
| OmniScene | 0.36 | 0.77 | 0.37 | 0.34 | 0.72 | 0.44 | 0.17 | 0.38 | 0.79 |
| Splatter360 | 0.47 | 0.91 | 0.28 | 0.40 | 0.71 | 0.35 | 0.17 | 0.40 | 0.75 |
| PanSplat | 0.41 | 1.08 | 0.11 | 0.40 | 1.07 | 0.13 | 0.39 | 1.06 | 0.14 |
| **Ours** | **0.20** | **0.48** | **0.75** | **0.17** | **0.41** | **0.80** | **0.11** | **0.30** | **0.89** |

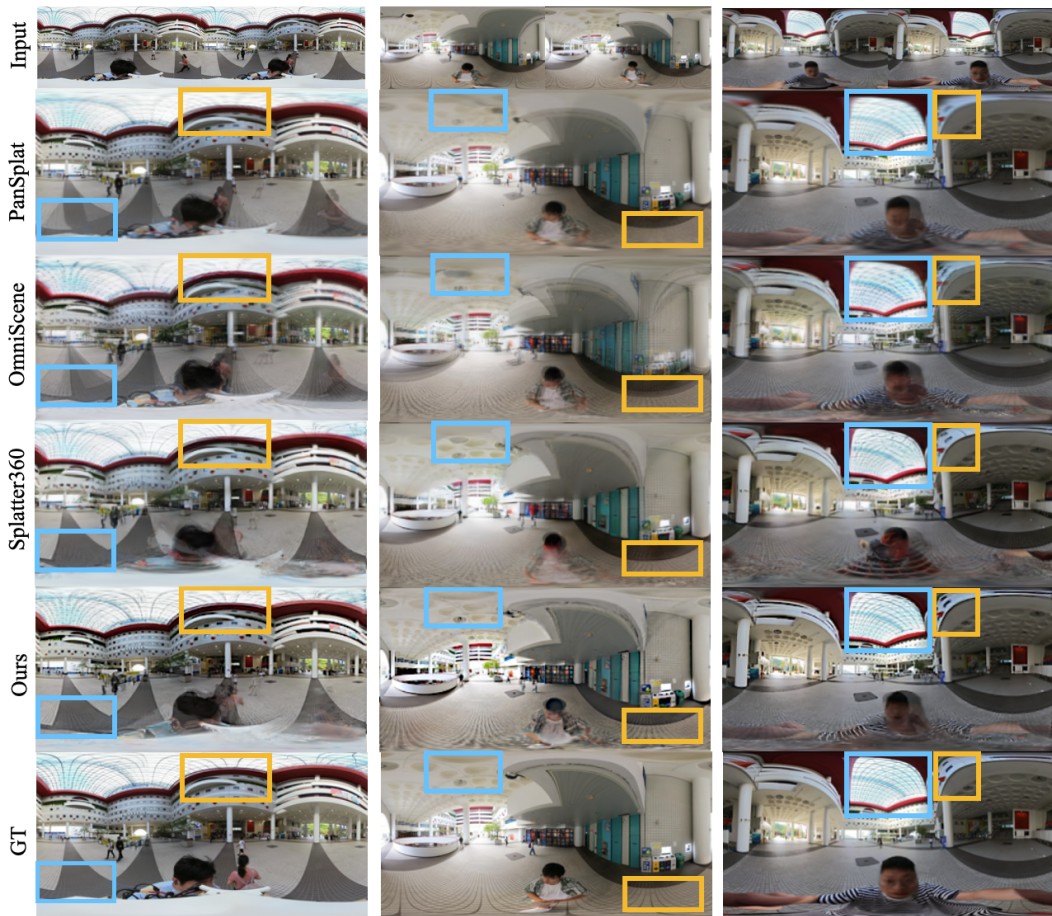

Figure 8: **Qualitative comparison on the real-world outdoor 360Loc dataset for the two-view input task with a 1.4m baseline.** We highlight specific regions, including ceilings, curved walls, and floors. As observed, our method maintains high rendering fidelity even in challenging non-Manhattan environments.

ter360, PanSplat) that require GT supervision, our method relies solely on RGB images. Outperforming fully supervised baselines demonstrates genuine geometric robustness.

## 5 CONCLUSION

In this work, we introduce CylinderSplat, a new feed-forward framework for panoramic 3DGS. Our method combines a flexible, attention-based pixel branch for high-fidelity reconstruction with a geometrically aware volume branch for robust completion of occluded regions. The main contribution of our cylindrical Triplane is a representation that offers a better fit for the unique geometry of 360° panoramic images than previous Cartesian or spherical approaches, enabling better handling of both synthetic and real-world environments with varying numbers of views as input. One area for future improvement is our fusion mechanism, which is currently a simple concatenation that may not always produce perfectly seamless completions. We will aim to develop more advanced fusion techniques to reduce redundant Gaussians to improve the quality of the fused reconstruction.

## REPRODUCIBILITY STATEMENT

The implementation details of our model are provided in Section 3, with training settings and evaluation protocols provided in Section 4 and Appendix B. Additional ablation studies are included in the 4.2 and Appendix B.2, B.1, B.3 to clarify the effect of individual components. We promise to release both the dataset and the code to facilitate reproducibility.

## ACKNOWLEDGE

The authors are grateful for the valuable comments and suggestions by the reviewers and ACs. This work was supported by NSFC (62406194), Shanghai Frontiers Science Center of Human-centered Artificial Intelligence (ShangHAI), MoE Key Laboratory of Intelligent Perception, HPC Platform of ShanghaiTech University and Human-Machine Collaboration (KLIP-HuMaCo). A part of the experiments of this work were supported by the core facility Platform of Computer Science and Communication, SIST, ShanghaiTech University.

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

## A STATEMENT ON THE USE OF LLM

The writing and polishing of this manuscript were greatly supported by the Large Language Model (LLM) Gemini 2.5 Pro. The LLM helped improve the language quality, including fixing grammar, refining sentence structure, rephrasing for clarity and conciseness, and ensuring consistent terminology and overall readability.

It is essential to clarify that the LLM's role was strictly limited to linguistic enhancement and did not involve any part of the scientific process. All intellectual and conceptual contributions—such as research design, methodology, data analysis, and result interpretation—are entirely credited to the human authors. After any AI-assisted editing, the authors carefully reviewed and manually revised the text to ensure accuracy and fidelity to their original intent. The authors take full responsibility for all content, scientific accuracy, and ethical standards of this manuscript.

## B IMPLEMENTATION DETAILS

**Training and Supervision.** To ensure broad applicability, we utilize depth maps from the UniK3D (Piccinelli et al., 2025) foundation model, which serve as both the ground truth for supervision and the input geometric prior for our pixel branch. Additionally, for calculating the PCC metric, we use depth from DepthAnywhere (Wang & Liu, 2024) as our reference ground-truth depth. All experiments were conducted on two NVIDIA RTX 4090 GPUs with a batch size of 2, and the input image resolution is consistently $512 \times 1024$. Our training process begins with the Matterport3D dataset, where we utilize our full three-stage curriculum. We first train the pixel branch, then the volume branch, and finally both branches jointly, with each stage lasting 10 epochs. Subsequently, for the 360Loc dataset, we fine-tune the model for an additional 10 epochs, specifically during the final joint-training stage. It is important to note that all other fine-tuning experiments mentioned in this paper also follow this joint-tuning protocol; the full three-stage curriculum is used only for the initial pre-training phase. For optimization, we use the AdamW (Loshchilov & Hutter, 2017) optimizer with a learning rate of $1 \times 10^{-4}$ and a cosine decay schedule.

**Triplane Configuration and Initialization.** Our cylindrical Triplane defaults to the configuration of $(N_r, N_z, N_\theta) = (16, 64, 128)$ and $(N'_r, N'_z, N'_\theta) = (8, 32, 64)$ for the $(r, z, \theta)$ axes, covering a 10m radius and height. A key aspect of our multi-Triplane strategy is the initialization process, which adapts to the scene's content. For synthetic scenes where a static scene dominates, we initialize each local Triplane by projecting feature points from *all* camera views that fall within its cylindrical volume. This serves to enrich the scene information within each Triplane. In contrast, for real-world scenes where dynamic objects like pedestrians and vehicles are unavoidable, we initialize each Triplane using *only* the feature points from the camera at its center. This avoids introducing noise from dynamic objects captured in other views.

**Direct Panoramic Rendering.** During the rendering stage, unlike prior feed-forward methods that render six cubemap faces and stitch them into a panorama, we employ a 3DGS rasterizer designed specifically for panoramas. This allows us to render the full equirectangular image in a single pass, significantly improving our rendering speed.

In the following sections, we provide a more detailed analysis of our choices regarding Triplane resolution, the initialization strategies, and our direct rendering approach.

### B.1 ABLATION ON TRIPLANE RESOLUTION

To justify the configuration of our cylindrical Triplane, we conducted a detailed ablation study on the sampling hyperparameters. Specifically, we examined the grid resolutions along the $(r, z, \theta)$ axes for two distinct stages: $(N_r, N_z, N_\theta)$, utilized during Cross-Plane Attention sampling, and $(N'_r, N'_z, N'_\theta)$, utilized during Triplane-to-Image Attention sampling. To strictly isolate the impact of the volumetric representation, we performed novel view synthesis using only the Volume Branch, while maintaining a constant physical bounding volume (10m radius, $360°$ azimuth, and 10m height).

We adapt a controlled variable approach to analyze the efficiency trade-offs: first, we fix $(N'_r, N'_z, N'_\theta)$ at $(8, 32, 64)$ while varying $(N_r, N_z, N_\theta)$; subsequently, we fix $(N_r, N_z, N_\theta)$ at

$(16, 64, 128)$ while varying $(N'_r, N'_z, N'_\theta)$. As presented in Table 8, the results demonstrate a clear trend: while increasing sampling density consistently improves reconstruction quality, it imposes a corresponding penalty on GPU memory usage and inference latency. Consequently, we selected the configuration of $(N_r, N_z, N_\theta) = (16, 64, 128)$ and $(N'_r, N'_z, N'_\theta) = (8, 32, 64)$ (highlighted in bold), as this combination strikes the optimal balance between performance fidelity and computational efficiency.

Table 8: Ablation study on sampling hyperparameters ($(N_r, N_z, N_\theta)$ and $(N'_r, N'_z, N'_\theta)$) on the Matterport3D (2.0m baseline) dataset with only the volume branch.

| Configuration $(r, z, \theta)$ | PCC ↑ | WS-PSNR ↑ | SSIM ↑ | LPIPS ↓ | Memory | Time |
|---|---|---|---|---|---|---|
| $(N_r, N_z, N_\theta) = (16, 32, 64)$ | 0.72 | 21.18 | 0.71 | 0.24 | 15GB | 0.26s |
| $(N_r, N_z, N_\theta) = (16, 64, 128)$ | **0.78** | **22.17** | **0.78** | **0.21** | **17GB** | **0.29s** |
| $(N_r, N_z, N_\theta) = (16, 128, 256)$ | 0.80 | 23.88 | 0.82 | 0.19 | 25GB | 0.34s |
| $(N'_r, N'_z, N'_\theta) = (8, 16, 32)$ | 0.77 | 22.08 | 0.73 | 0.28 | 16GB | 0.27s |
| $(N'_r, N'_z, N'_\theta) = (8, 32, 64)$ | **0.78** | **22.17** | **0.78** | **0.21** | **17GB** | **0.29s** |
| $(N'_r, N'_z, N'_\theta) = (8, 64, 128)$ | 0.79 | 22.46 | 0.80 | 0.21 | 18GB | 0.31s |

## B.2 TRIPLANE INITIALIZATION STRATEGIES

For the distinct domains of static and dynamic scene reconstruction, we employ two different Triplane initialization strategies, as illustrated in Fig. 9. The first strategy, for static scenes, maximizes information by aggregating feature points from all camera views. The second, for dynamic scenes, mitigates noise by using features only from the corresponding camera view. We conducted a comparative experiment (Table 9) to validate this design choice. The results confirm that the all-view aggregation strategy (Fig. 9(a)) is optimal for datasets composed primarily of static scenes (e.g., Matterport3D), while the single-view strategy (Fig. 9(b)) is superior for datasets with dynamic elements (e.g., 360Loc).

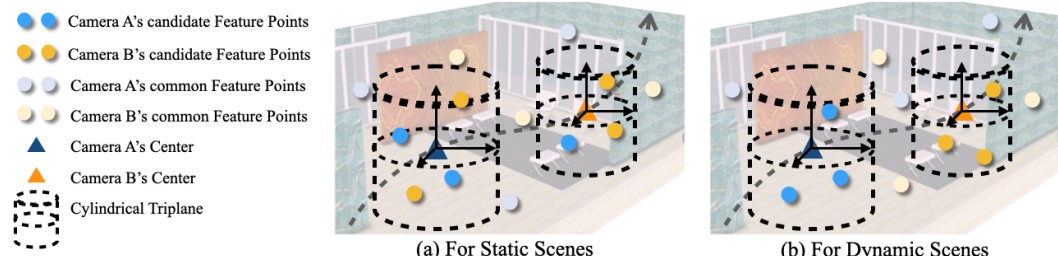

(a) For Static Scenes      (b) For Dynamic Scenes

Figure 9: **Visualization of our Triplane initialization strategies.** In this figure, a blue triangle represents Camera A, with blue circles denoting its corresponding feature point cloud. A yellow triangle represents Camera B with its yellow point cloud. We construct a local cylindrical Triplane at each camera's location and initialize it by projecting feature points onto its three planes. **(a)** For static scenes, the Triplane for Camera A is formed by aggregating all candidate feature points that fall within its volume, regardless of their origin (both blue and yellow points). **(b)** In contrast, for scenes with dynamic objects, the Triplane for Camera A is formed exclusively by feature points from its own view (only blue points). This strategy prevents dynamic inconsistencies from one view (e.g., yellow points from Camera B) from corrupting another view's Triplane with noise.

Table 9: Comparison of different Triplane initialization strategies on static (Matterport3D) and dynamic (360Loc) datasets.

| Dataset | Matterport3D (2.0m baseline) | | | | 360Loc (avg. 1.40m baseline) | | | |
|---|---|---|---|---|---|---|---|---|
| Method | PCC↑ | WS-PSNR↑ | SSIM↑ | LPIPS↓ | PCC↑ | WS-PSNR↑ | SSIM↑ | LPIPS↓ |
| Shared Initialization | **0.851** | **23.76** | **0.835** | **0.175** | 0.831 | 27.33 | 0.841 | 0.107 |
| Isolated Initialization | 0.847 | 23.24 | 0.796 | 0.197 | **0.883** | **28.35** | **0.896** | **0.095** |

### B.3 COMPARISON OF 3DGS RENDERING METHODS

Prior feed-forward methods for panoramic NVS typically rely on an indirect rendering pipeline: they first render six perspective views to form a cubemap, which is then stitched together to create the final panoramic image. In contrast, our method utilizes a 3DGS rasterizer (Li et al., 2025), designed explicitly for panoramas, which enables us to render the full equirectangular image directly. This approach is more efficient and requires two core modifications to the standard splatting pipeline.

First, to project the 3D Gaussians into the 2D panoramic image space, we replace the standard pinhole projection with an equirectangular projection function. For a 3D Gaussian centered at $(x, y, z)$, its corresponding 2D pixel coordinate $(u, v)$ in a panorama of resolution $H \times W$ is calculated as:

$$u = \frac{W}{2\pi} \left( \text{atan2}(x, z) + \pi \right), \tag{8}$$

$$v = \frac{H}{\pi} \left( \text{atan2}(y, \sqrt{x^2 + z^2}) + \frac{\pi}{2} \right). \tag{9}$$

Second, the Jacobian of this projection function, which is used to transform the 3D Gaussian covariances into 2D, must be updated. The general form of the Jacobian $\mathbf{J}_i$ for a Gaussian $i$ is the matrix of partial derivatives of the pixel coordinates $(u_i, v_i)$ with respect to the 3D world coordinates $(x, y, z)$:

$$\mathbf{J}_i = \begin{bmatrix} \dfrac{\partial u_i}{\partial x} & \dfrac{\partial u_i}{\partial y} & \dfrac{\partial u_i}{\partial z} \\ \dfrac{\partial v_i}{\partial x} & \dfrac{\partial v_i}{\partial y} & \dfrac{\partial v_i}{\partial z} \end{bmatrix}. \tag{10}$$

For our specific panoramic projection, this results in the following specialized Jacobian:

$$\mathbf{J}_i = \begin{bmatrix} \dfrac{W}{2\pi} \cdot \dfrac{z_i}{x_i^2 + z_i^2} & 0 & -\dfrac{W}{2\pi} \cdot \dfrac{x_i}{x_i^2 + z_i^2} \\ \dfrac{H}{\pi} \cdot \dfrac{x_i y_i}{r_i^2 \sqrt{x_i^2 + z_i^2}} & \dfrac{H}{\pi} \cdot \dfrac{\sqrt{x_i^2 + z_i^2}}{r_i^2} & -\dfrac{H}{\pi} \cdot \dfrac{z_i y_i}{r_i^2 \sqrt{x_i^2 + z_i^2}} \end{bmatrix}, \tag{11}$$

where $r_i = \sqrt{x_i^2 + y_i^2 + z_i^2}$. Using these updated calculations allows us to render panoramic views directly. Furthermore, as validated by the experiment in Table 10, while the rendering quality of our direct panoramic 3DGS is comparable to the indirect method of stitching six pinhole views (cubemaps), our direct approach is faster.

Table 10: Comparison of different rendering methods on the Matterport3D dataset for the two-view input task, evaluating quality metrics and inference time.

| Dataset | Matterport3D | | | | | | | | | | | | |
|---|---|---|---|---|---|---|---|---|---|---|---|---|---|
| Baseline | 2.0m | | | | 1.5m | | | | 1.0m | | | | Inference Time |
| Render Method | PCC↑ | WS-PSNR↑ | SSIM↑ | LPIPS↓ | PCC↑ | WS-PSNR↑ | SSIM↑ | LPIPS↓ | PCC↑ | WS-PSNR↑ | SSIM↑ | LPIPS↓ | |
| CubeMap | 0.815 | 23.71 | 0.822 | 0.171 | 0.853 | 25.51 | 0.856 | 0.129 | 0.906 | 28.86 | 0.936 | 0.076 | 0.33s |
| Panorama Gaussian | 0.851 | 23.76 | 0.835 | 0.175 | 0.867 | 25.91 | 0.873 | 0.128 | 0.923 | 28.89 | 0.937 | 0.081 | 0.29s |

## C MOTIVATION FOR CYLINDRICAL TRIPLANES

As illustrated in Fig. 10, our cylindrical Triplane is motivated by four key advantages over Cartesian and spherical alternatives.

**Cartesian Triplane.** Even though a flat, box-like Cartesian plane (as in Fig. 10(a)) works well for structured environments like buildings with straight walls, it introduces significant problems when dealing with 360-degree panoramic images. When information from a flat grid is projected onto a panorama, the image becomes stretched and blurry. The ability of this plane to represent details accurately quickly diminishes when the plane is too close or too far from the camera. Points on the same Cartesian plane, when projected onto a panoramic image, suffer from severe distortion (see Fig. 10(c)). Furthermore, when mapping 360-degree information back onto these flat planes, the

Cartesian system's lack of omnidirectional awareness leads to many feature points from different panoramic pixels projecting onto the same grid cells of the Triplane. This results in messy and unevenly distributed information on the plane (see Fig. 10(b)).

**Spherical Triplane.** A spherical coordinate system naturally aligns with equirectangular pixel grids, offering theoretically perfect uniform sampling where every feature corresponds to an equal portion of the $360°$ view (as shown in Fig. 10(b,c)). However, this geometric elegance poorly fits real-world "Manhattan-world" scenes (Fig. 10(a)). Spherical Triplanes struggle to represent flat surfaces, such as floors and ceilings, and exhibit severe distortion at their poles.

**Cylindrical Triplane.** The cylindrical coordinate system provides a compelling balance. As shown in Fig. 10(a), it effectively models Manhattan-world structures (floors, ceilings, walls) without the severe distortions produced by the spherical system. Its sampling pattern (Fig. 10(c)), while not perfectly uniform, is far more stable than Cartesian. This distribution beneficially prioritizes top/bottom features at small radii and central panoramic regions at large radii, which aligns well with real-world panoramic depth distributions, where the central region typically exhibits the greater depth and the top/bottom regions the smaller. Furthermore, projecting panoramic feature clouds onto a cylindrical Triplane results in a relatively uniform feature distribution (Fig. 10(b)).

Finally, qualitative rendering comparisons using only the volume branch (Fig. 10(d)) demonstrate that our cylindrical Triplane consistently produces novel views with significantly more detail and fewer distortion artifacts than its Cartesian and spherical counterparts.

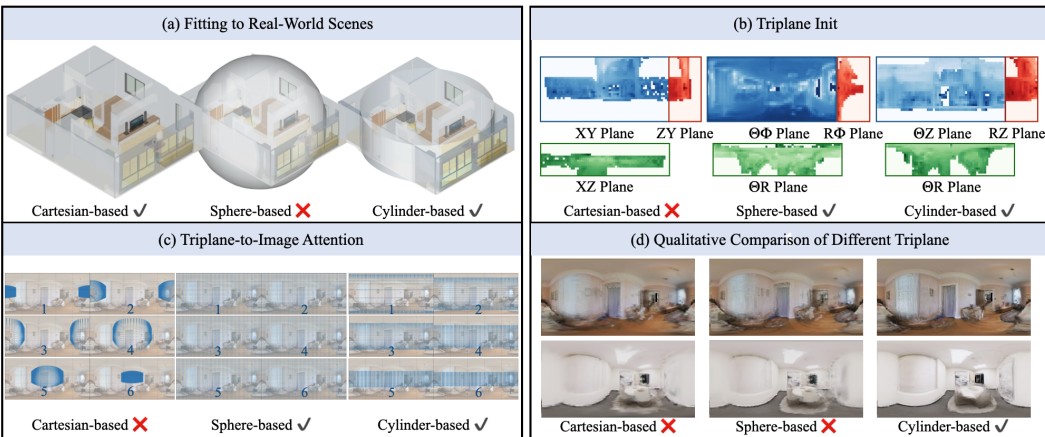

Figure 10: **Visualizing the advantages of the cylindrical Triplane. (a)** The shape of the sampling volumes for each coordinate system when fitting a typical synthetic scene, highlighting their geometric alignment. **(b)** Feature distribution during Triplane initialization. At this stage, panoramic feature point clouds are projected onto the Triplane surfaces. The Cartesian system suffers from heavy point overlap, where many distinct 3D feature points are projected to the same grid cells. In contrast, the spherical and cylindrical systems achieve a much more even distribution of features. **(c)** Projection patterns for Triplane-to-Image Attention. We visualize how sample points from each Triplane's volume project onto the panorama for cross-attention. This is shown across six maps for each coordinate system, ordered by increasing distance from the origin. For the Cartesian system (left), maps show that on a Cartesian plane, as this plane moves farther from the origin, its panoramic projection exhibits a limited field of view and severe distortion. For the spherical system (middle), maps illustrate that on a spherical surface, with increasing sphere radius (distance from origin), its projection remains consistently uniform across the panorama. Finally, for the cylindrical system (right), maps display that on a cylindrical surface, as its radius increases, the projection remains relatively uniform but naturally focuses more on the panoramic center, better aligning with real-world depth distributions. **(d)** Qualitative rendering comparison using only the volume branch, demonstrating the superior performance of the cylindrical representation in novel view synthesis.

# D  ABLATE DIFFERENT DEPTH PRIORS (E.G., DEPTHANYWHERE, ZOEDEPTH)

To demonstrate the robustness of our framework, we substitute our default UniK3D (Piccinelli et al., 2025) prior with DepthAnywhere (Wang & Liu, 2024) and UniFuse (Jiang et al., 2021). We specifically include UniFuse as it serves as the depth prior for both PanSplat and Splatter360.

Table 11 demonstrates that our method maintains consistent performance advantages across different depth priors. Even when utilizing the exact same UniFuse prior as PanSplat and Splatter360, our approach yields superior quality. Crucially, we achieve these results without ground truth (GT) depth supervision; in contrast, both Splatter360 and PanSplat rely on UniFuse for priors and additionally require GT depth as supervision during training.

Table 11: Different depth prior on Matterport3D 2.0m baseline.

| Method | AbsRel ↓ | RMSE ↓ | $\delta_1$ ↑ | PCC ↑ | PSNR ↑ | SSIM ↑ | LPIPS ↓ |
|---|---|---|---|---|---|---|---|
| OmniScene | 0.36 | 0.77 | 0.37 | 0.73 | 22.75 | 0.71 | 0.24 |
| Splatter360 | 0.47 | 0.91 | 0.28 | 0.68 | 21.31 | 0.74 | 0.28 |
| PanSplat | 0.41 | 1.08 | 0.11 | 0.71 | 20.56 | 0.77 | 0.26 |
| **Ours (DepthAnywhere)** | **0.34** | **0.66** | **0.60** | **0.78** | **23.08** | **0.78** | **0.22** |
| **Ours (UniFuse)** | **0.31** | **0.64** | **0.62** | **0.78** | **23.31** | **0.81** | **0.21** |
| **Ours (UniK3D)** | **0.20** | **0.48** | **0.75** | **0.85** | **23.76** | **0.83** | **0.17** |

# E  DERIVATION AND VALIDITY OF SCALE TRANSFORMATION ($S'$)

**1. Rotation ($R$).**  First, we clarify that Rotation ($R$) does not undergo coordinate transformation. Consistent with standard 3DGS, our MLP predicts rotation (as quaternions) directly in the global Cartesian coordinate system. The coordinate transformation (Eq. 3 & 4) applies only to the local position offset ($\delta_{\text{local}}$) and scale ($S_{\text{local}}$).

**2. Local Position ($\delta_{\text{local}}$) and Scale ($S_{\text{local}}$).**  Although $S'$ involves a local approximation, this design strictly follows the standard "volume-bounded" optimization principle used in existing grid-based 3DGS methods. As illustrated in Fig. 2(d) of our paper, methods like OmniScene constrain each Gaussian primitive within a fixed volume element ("Cartesian Volume" in Fig. 2(d)) defined by the Triplane grid resolution (e.g., dimensions of $2\delta_x, 2\delta_y, 2\delta_z$). The learnable position offsets and scales are constrained within this grid (e.g., $\pm\delta_x, \pm\delta_y, \pm\delta_z$). This constraint is critical for ensuring training stability.

We apply this same principle to our cylindrical representation. As shown in the "Cylindrical Volume" of Fig. 2(d), our grid units are curvilinear frustums defined by $(2\delta_r, 2\delta_\theta, 2\delta_z)$. Consequently, our network predicts local offsets and scales ($\delta_{\text{local}}, S_{\text{local}}$) restricted within these local bounds (e.g., $\pm\delta_r, \pm\delta_\theta, \pm\delta_z$). Since the standard 3DGS rasterizer accepts only Cartesian inputs, we must transform these locally predicted parameters into the global Cartesian frame. This necessitates the coordinate transformation for positions (Eq. 3) and the Jacobian-based transformation for scales (Eq. 4).

Finally, we perform an additional ablation study on Matterport3D (2.0m baseline) comparing our Jacobian-based scaling with predicting Cartesian scales directly within the cylindrical triplane. The results confirm that respecting the local cylindrical geometry through our transformation outperforms the alternative.

Table 12: Ablation study on scale transformation strategy (Matterport3D, 2.0m baseline).

| Method | abs_rel ↓ | rmse ↓ | delta1 ↑ | PCC ↑ | PSNR ↑ | SSIM ↑ | LPIPS ↓ |
|---|---|---|---|---|---|---|---|
| Ours (Cartesian Scale) | 0.22 | 0.55 | 0.73 | 0.82 | 23.54 | 0.82 | 0.18 |
| **Ours (Cylinder Scale)** | **0.20** | **0.48** | **0.75** | **0.85** | **23.76** | **0.84** | **0.18** |

## F PERFORMANCE ON WIDE-BASELINE REAL-WORLD SCENARIOS

To substantiate our SOTA claim in real-world settings, we conducted two additional evaluations focusing on challenging wide-baseline scenarios where geometric completion is most critical:

1. 360Loc with wider baselines (3.0 m and 4.5 m).

2. **A new large-scale real-world dataset curated from Google Street View (Kansas City), comprising 8,500 sequences with extreme baselines (20–35 m) and fewer dynamic objects.**

As reported in Tables 13, although the gains at a short baseline (1.4 m) are moderate, our advantage increases substantially as the baseline widens. On the Kansas dataset (20 m+ baseline), we outperform the strongest competitor (OmniScene) by +3.95 dB in WS-PSNR. This confirms that CylinderSplat excels in sparse-view synthesis under challenging real-world conditions.

Table 13: **Quantitative comparison on wide-baseline real-world scenarios.** We report results on the Kansas dataset (extreme 20m–30m baseline) and 360Loc dataset with increasing baselines (4.5m and 3.0m). Our method consistently outperforms baselines, with the performance gap widening as the difficulty increases.

| Method | Kansas Dataset (20m–30m) | | | | 360Loc (4.5m Baseline) | | | | 360Loc (3.0m Baseline) | | | |
|---|---|---|---|---|---|---|---|---|---|---|---|---|
| | PCC ↑ | PSNR ↑ | SSIM ↑ | LPIPS ↓ | PCC ↑ | PSNR ↑ | SSIM ↑ | LPIPS ↓ | PCC ↑ | PSNR ↑ | SSIM ↑ | LPIPS ↓ |
| OmniScene | 0.55 | 17.08 | 0.44 | 0.35 | 0.84 | 18.12 | 0.53 | 0.41 | **0.86** | 21.83 | 0.66 | 0.30 |
| Splatter360 | 0.39 | 16.21 | 0.36 | 0.47 | 0.80 | 19.93 | 0.62 | 0.32 | 0.82 | 22.65 | 0.71 | 0.24 |
| PanSplat | 0.07 | 15.02 | 0.29 | 0.61 | 0.10 | 21.03 | 0.69 | 0.28 | 0.11 | 23.47 | 0.75 | 0.22 |
| **Ours** | **0.63** | **21.03** | **0.67** | **0.21** | **0.86** | **22.68** | **0.73** | **0.22** | **0.86** | **24.50** | **0.79** | **0.18** |

## G TRAINING COMPLEXITY AND EFFICIENCY ANALYSIS

We clarify that the three-stage training is required only for the initial training on the Matterport3D dataset to ensure robust initialization of the independent Pixel and Volume branches. For all other datasets (e.g., 360Loc, Kansas), we only train the third stage (Joint Training), fine-tuning from the weights trained on Matterport3D.

As shown in Table 14:

1. **On Matterport3D:** Even with separate stages, the combined per-iteration time of Stage 1 and Stage 2 ($0.81s + 1.13s = 1.94s$) is still faster than a single iteration of PanSplat (2.17s). We need 10 more epochs for the final joint stage.

2. **On Other Datasets (e.g., 360Loc):** Since we only execute the third stage, our training time per iteration (1.98s) is faster than all competing methods (PanSplat 2.17s, Splatter360 2.89s, OmniScene 3.23s).

Table 14: **Comparison of training efficiency (time per iteration and epochs) against baselines.**

| Method | Training Time (per iteration) | Training Epochs |
|---|---|---|
| OmniScene | 3.23s | 10 |
| Splatter360 | 2.89s | 10 |
| PanSplat | 2.17s | 10 |
| **Ours (Stage 1: Pixel Branch)** | **0.81s** | **10** |
| **Ours (Stage 2: Volume Branch)** | **1.13s** | **10** |
| **Ours (Stage 3: Joint Training)** | **1.98s** | **10** |

## H ANALYSIS OF PANORAMIC ARTIFACTS: SEAMS AND POLES

1. **Seams**: The Cylindrical Triplane is logically circular in the $\theta$ dimension; queries at $\theta = 0$ and $\theta = 2\pi$ access the exact same physical location in the coordinate system, rendering our

method naturally seamless. To quantify this, we evaluate the Left-Right Consistency Error (LRCE) (the mean pixel difference between the left and right boundaries). As shown in Table 15, our method achieves an error that is an order of magnitude lower than competing methods (e.g., 0.025 vs. PanSplat's 0.088), confirming superior continuity.

2. **Poles**: As visualized in Figure 10(c) of the supplementary material, our Cylindrical Triplane leverages a geometric prior that optimally aligns with real-world depth distributions: it prioritizes sampling at the poles for regions with small radii (close to the camera) while focusing on the central horizon for regions with large radii (far from the camera). Furthermore, by maintaining uniform resolution along the $Z$-axis, our representation ensures consistent detail for the zenith and nadir regions, allowing the geometric representation itself to robustly handle polar areas as shown in (Fig. 4, Fig. 5, Fig. 6, and Fig 8).

Table 15: **Left-Right Consistency Error (LRCE)** ↓. Lower values indicate better continuity at panoramic boundaries.

| Method | Mp3D (2.0m) | Mp3D (1.5m) | Mp3D (1.0m) |
|---|---|---|---|
| OmniScene | 0.291 | 0.325 | 0.299 |
| Splatter360 | 0.165 | 0.146 | 0.128 |
| PanSplat | 0.137 | 0.119 | 0.088 |
| **Ours** | **0.025** | **0.024** | **0.023** |

Table 16: Quantitative comparison of advanced Gaussian fusion strategies on the Matterport3D dataset (2.0m baseline).

| Method | PCC ↑ | WS-PSNR ↑ | SSIM ↑ | LPIPS ↓ |
|---|---|---|---|---|
| Density Clustering | 0.84 | 22.52 | 0.75 | 0.24 |
| **Depth Pruning** | **0.87** | **23.91** | **0.87** | **0.15** |
| Ours (Original) | 0.85 | 23.76 | 0.83 | 0.17 |

# I  ANALYSIS OF ADVANCED GAUSSIAN FUSION STRATEGIES

To explore fusion strategies beyond simple concatenation, we implemented and evaluated both density clustering and depth-guided pruning.

As shown in Table 16, our findings are as follows:

- **Density Clustering:** We attempted to merge tightly packed Gaussians by learning confidence weights via a softmax mechanism. However, we found that the network struggled to learn reliable confidence scores for fusion, leading to visual artifacts (distortions and holes) and a performance drop compared to our original concatenation baseline.

- **Depth-Guided Pruning:** Conversely, this strategy proved effective. By reducing the opacity of Gaussians that deviate significantly from the predicted depth and pruning low-confidence primitives, we achieved improvements in both geometric accuracy and rendering quality.

# J  RE-EVALUATING OMNISCENE WITH DIRECT PANORAMIC RASTERIZER.

To isolate the impact of the rendering pipeline from the geometric representation, we re-evaluate OmniScene using our direct panoramic rasterizer. As shown in Table 17, replacing the cubemap renderer with our direct rasterizer yields comparable overall performance. This mirrors the conclusion in Table 10, where we demonstrated that switching CylinderSplat to a cubemap renderer had negligible impact on quality. Thus, we conclude that the direct rasterizer primarily contributes to inference speed, while the superior reconstruction quality is driven by our Cylindrical Triplane representation.

Table 17: Comparison of OmniScene performance using Cubemap vs. Direct Panoramic Rasterizer on Matterport3D 2.0m baseline.

| Method | AbsRel ↓ | RMSE ↓ | $\delta_1$ ↑ | PCC ↑ | PSNR ↑ | SSIM ↑ | LPIPS ↓ | Inference Time |
|---|---|---|---|---|---|---|---|---|
| OmniScene (Cubemap) | 0.37 | 0.65 | 0.53 | 0.73 | 22.75 | 0.71 | 0.24 | 0.48s |
| OmniScene (Direct Raster) | 0.29 | 0.63 | 0.61 | 0.76 | 22.19 | 0.76 | 0.24 | 0.44s |

## K    LIMITATIONS

Our method encounters challenges in parts of the 360Loc dataset, where unavoidable dynamic elements (e.g., the photographer at the nadir, moving pedestrians). Since we do not explicitly optimize for transient objects, these can cause ghosting artifacts in the synthesized views. We have visualized these specific cases in the Fig 8, noting that while our method exhibits ghosting, this issue is equally prevalent in competing methods.

## L    FURTHER SCENE VISUALIZATIONS

We provide additional scene visualizations by comparing our method with others on both two-view and single-view tasks, examining rendered RGB images and depth maps as shown in Fig. 11 and Fig. 12. Compared to Triplane-based methods like OmniScene, which utilizes a Cartesian Triplane, our approach demonstrates superior performance. OmniScene often produces noticeable artifacts, distortions, and loss of detail in rendered RGB and depth maps, particularly near the bottom of the panorama (e.g., ground regions). Specifically, its depth maps frequently exhibit striped artifacts near the ceiling and floor. Furthermore, compared to CostVolume-based methods (including Splatter360 and PanSplat), our approach avoids the common issues of holes and distortions that arise when input view spacing is large. Our method excels at minimizing distortion while effectively filling in the occluded regions, offering a more complete and accurate reconstruction.

## M    VISUALIZATIONS OF NON-MANHATTAN AND OUTDOOR SCENES

We include additional visualizations in the supplementary material, specifically targeting outdoor scenes from the 360Loc dataset (Fig. 8) and complex urban/forest environments from the Kansas dataset (Fig. 13). These qualitative results demonstrate that even in scenarios featuring curved structures or unstructured outdoor geometry, our method maintains robust performance comparable to state-of-the-art baselines.

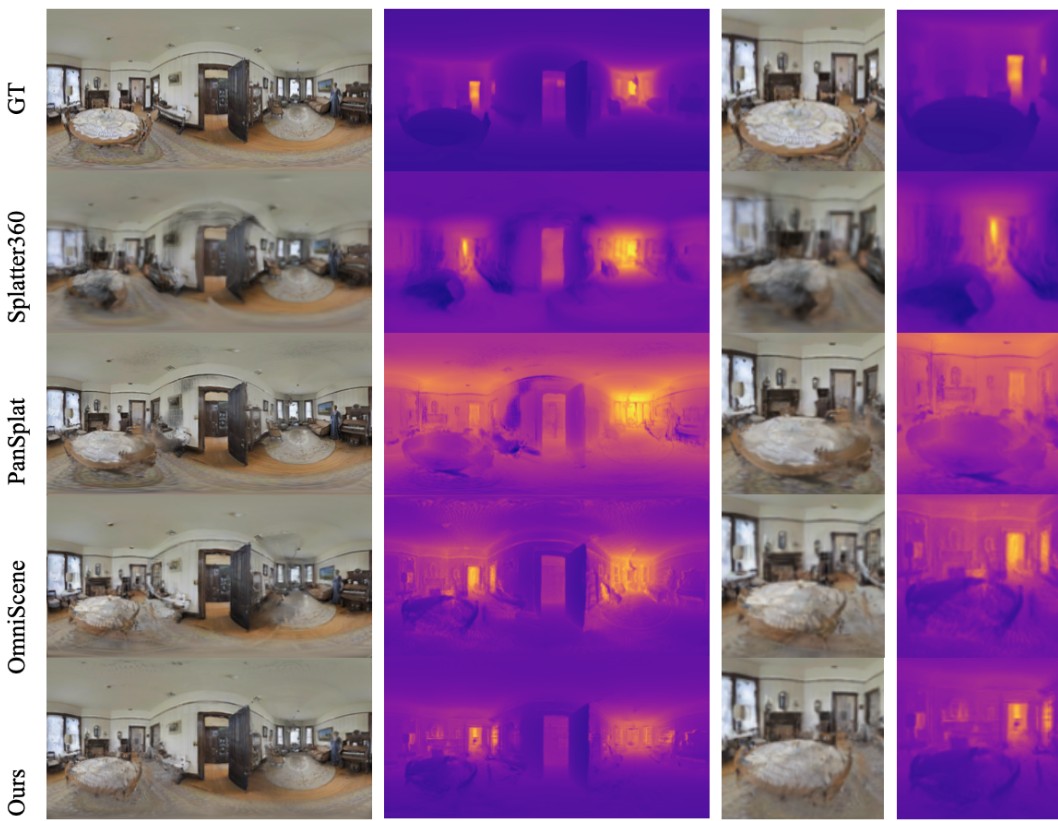

Figure 11: **Qualitative comparisons on synthetic datasets for the two-view input task,** with an input view baseline of 2.0m. The first column shows the rendered RGB image for a novel view, and the second column displays its corresponding depth map. The third and fourth columns provide zoomed-in results of the RGB image and depth map, respectively.

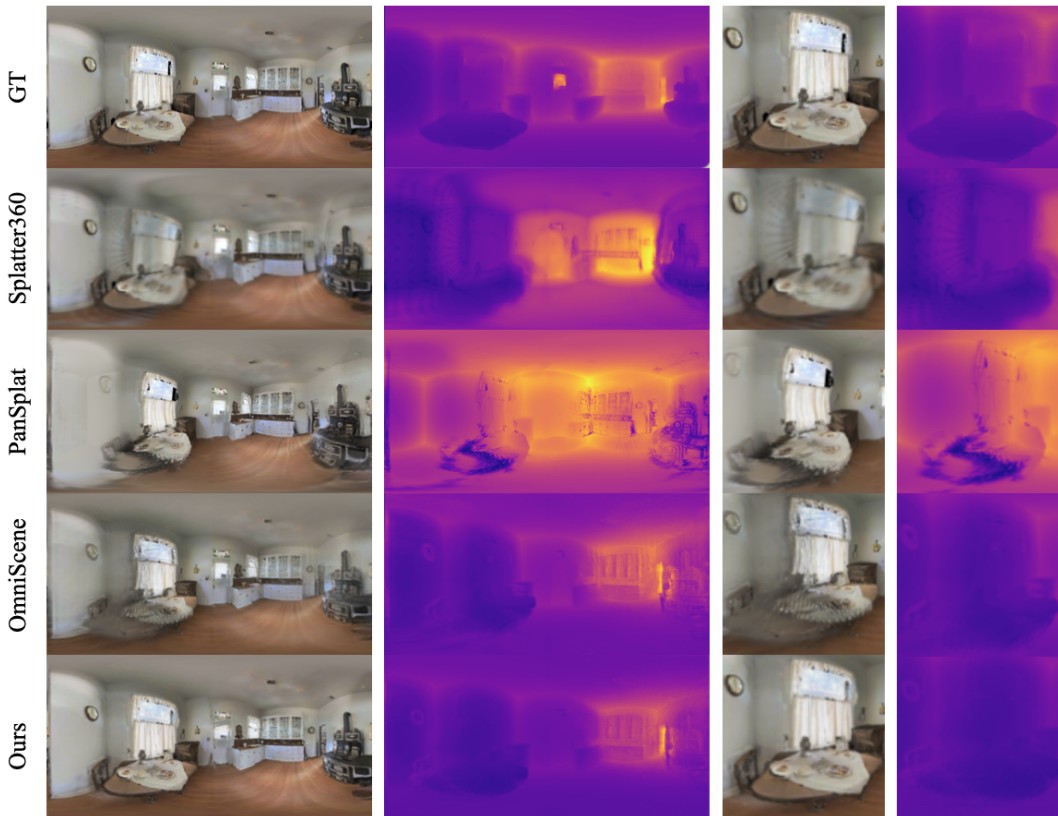

Figure 12: **Qualitative comparisons on synthetic datasets for the single-view input task,** where the distance between the input and output views is 1.0m. The first column shows the rendered RGB image for a novel view, and the second column displays its corresponding depth map. The third and fourth columns provide zoomed-in results of the RGB image and depth map, respectively.

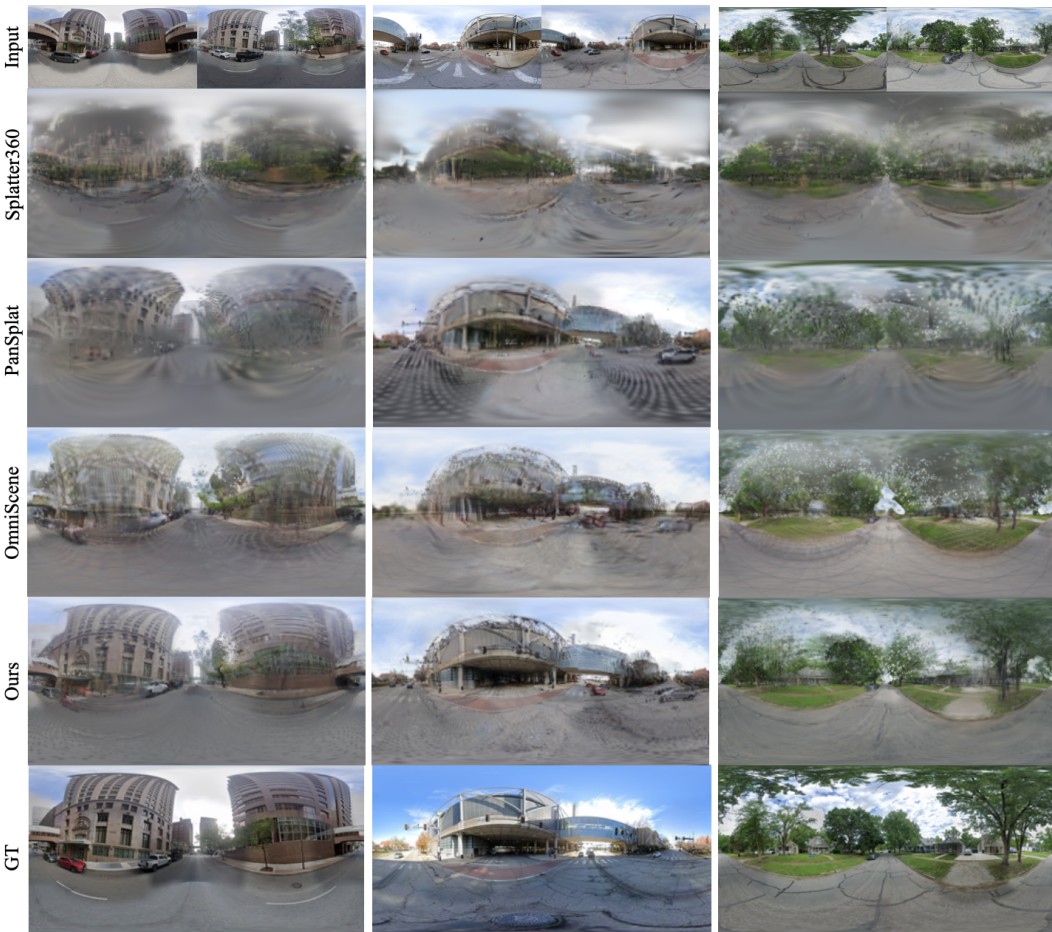

Figure 13: **Qualitative comparison on the real-world outdoor Kansas Dataset for the two-view input task.** The distance between the two input views is approximately 20–30m, making it a highly challenging scenario. Although some blurriness remains, our method significantly outperforms the baselines, demonstrating the effectiveness of CylinderSplat for sparse-view synthesis under demanding real-world conditions.

