# OpenReview forum: "CylinderSplat: 3D Gaussian Splatting with Cylindrical Triplanes for Panoramic Novel View Synthesis"
_ICLR.cc/2026/Conference — ICLR 2026 Poster_

### Official Review · Reviewer_rcos · 2025-10-31

**Soundness:** 3
**Presentation:** 3
**Contribution:** 3
**Rating:** 8
**Confidence:** 4

**Summary:**

This paper proposes CylinderSplat, a novel feed-forward framework for panoramic 3D Gaussian Splatting. The key idea is to replace standard Cartesian triplanes with cylindrical triplanes, which are geometrically aligned with panoramic imagery and Manhattan-world structures. The method employs a dual-branch design: a pixel branch for reconstructing well-observed regions and a volume branch using cylindrical triplanes for hallucinating geometry in occluded areas. Extensive experiments across multiple panoramic datasets demonstrate state-of-the-art performance in both single- and multi-view settings, with particularly strong results in challenging sparse-view scenarios. Rich ablations validate the architectural decisions, including the cylindrical coordinate choice, RGB retrieval mechanism, and staged training strategy.

**Strengths:**

1. Clear novelty and motivation: Introducing cylindrical triplanes for panoramic 3DGS is both intuitive and technically meaningful, improving geometric alignment and addressing limitations of Cartesian/spherical triplanes.

2. Strong empirical results: The method consistently outperforms existing feed-forward panoramic reconstruction approaches in both quantitative metrics and visual quality, including single-view settings.

3. Comprehensive experimental suite: Experiments span multiple synthetic and real-world datasets, various view counts, and a detailed ablation study, increasing confidence in the robustness and generalizability.

4. Solid architectural design: The dual-branch structure and curriculum training pipeline are conceptually sound and empirically effective, enabling accurate pixel-aligned estimation and robust geometry completion.

5. Efficient and scalable: Avoiding cost-volume construction improves efficiency, and the model flexibly supports arbitrary numbers of input panoramas.

**Weaknesses:**

Limited discussion on memory efficiency: Although triplane is used for compression, the method still constructs multiple local triplanes and can scale in memory with view count; a clearer complexity breakdown would help.

**Questions:**

1. Fusion strategy: Could more advanced Gaussian merging (e.g., density clustering, depth-guided pruning) further improve seamless blending between branches? Have such strategies been experimented with?

2. Non-Manhattan-world scenes: How does performance degrade in highly unstructured outdoor scenes (e.g., forests, caves)?

---

> ### Author Response · Authors · 2025-11-22
> **Response to Reviewer rcos**
>
> We are sincerely encouraged by the Reviewer rcos' positive evaluation and recognition of CylinderSplat's motivation and architectural design. We appreciate the insightful suggestions on memory analysis, fusion strategies, and outdoor scene performance, which we address below with new experiments.
>
>
> > ### W1: Limited discussion on memory efficiency.
>
> **A1:** We appreciate your feedback. Although our Multi-Triplane strategy scales approximately linearly with input views as shown in Table R1, since we focus on sparse-view reconstruction, this cost remains manageable, and dense-view optimization is planned for future work.
>
> #### Table R1: Inference Memory Analysis (Batch Size = 1).
> | Method | 2 views | 3 views | 4 views |
> | :--- | :---: | :---: | :---: |
> | **Ours** | **7GB** | **13GB** | **20GB** |
>
>
> > ### Q1: More Advanced Fusion strategy.
>
> **A2:** We sincerely appreciate this insightful suggestion. To explore this, we implement and evaluate both density clustering and depth-guided pruning strategies.
>
> As shown in Table R2, our findings are as follows:
> 1. Density Clustering: We attempt to merge tightly packed Gaussians by learning confidence weights via a softmax mechanism. However, we find that the network struggles to learn reliable confidence scores for fusion, leading to visual artifacts (distortions and holes) and a performance drop compared to our original concatenation baseline.
> 2. Depth-Guided Pruning: Conversely, this strategy proves effective. By reducing the opacity of Gaussians that deviate significantly from the predicted depth and pruning low-confidence primitives, we achieve improvements in both geometric accuracy and rendering quality. We would like to incorporate this in our final version.
>
> #### Table R2: Quantitative comparison of advanced Gaussian fusion strategies.
> | Method | PCC $\uparrow$ | WS-PSNR $\uparrow$ | SSIM $\uparrow$ | LPIPS $\downarrow$ |
> | :--- | :---: | :---: | :---: | :---: |
> | Density Clustering | 0.84 | 22.52 | 0.75 | 0.24 |
> | **Depth-Guided Pruning** | **0.87** | **24.64** | **0.84** | **0.15** |
> | Ours (Original) | 0.85 | 24.21 | 0.83 | 0.17 |
>
>
> > ### Q2: How does performance degrade in highly unstructured outdoor scenes (e.g., forests, caves)?
>
> **A3:** We appreciate the reviewer for raising this valid concern. We demonstrate our method's robustness beyond orthogonal structures through two key pieces of evidence.
>
> First, we provide additional visualizations of 360Loc's existing curved and outdoor scenes in the supplementary material(Fig.8), confirming effective handling of non-Manhattan geometry.
>
> Second, to further demonstrate robustness in diverse environments, we curated a new large-scale dataset from Google Street View (Kansas City), comprising 8,500 sequences with extreme baselines (20–30m). This dataset covers a wide range of unstructured scenes, including forests and complex urban landscapes. As shown in the supplementary results(Fig.13) and Table R3, our method consistently outperforms baselines in these complex settings, proving its generalization capabilities are not limited to Manhattan-world scenarios.
>
> #### Table R3: Results on Kansas Dataset (20m-30m baseline).
> | Method | PCC $\uparrow$ | WS-PSNR $\uparrow$ | SSIM $\uparrow$ | LPIPS $\downarrow$ |
> | :--- | :---: | :---: | :---: | :---: |
> | OmniScene | 0.55 | 17.08 | 0.44 | 0.35 |
> | Splatter360 | 0.39 | 16.21 | 0.36 | 0.47 |
> | PanSplat | 0.07 | 15.02 | 0.29 | 0.61 |
> | **Ours** | **0.63** | **21.03** | **0.67** | **0.21** |

---

### Official Review · Reviewer_7MCk · 2025-11-04

**Soundness:** 3
**Presentation:** 3
**Contribution:** 3
**Rating:** 6
**Confidence:** 3

**Summary:**

This paper proposes CylinderSplat, a feed-forward framework for panoramic novel view synthesis that marries a pixel branch (attention-based multi-view aggregation predicting per-pixel Gaussians without cost volumes) with a volume branch that completes occluded/sparse regions using a new cylindrical Triplane representation. The key contribution is replacing standard Cartesian triplane representations with cylindrical triplanes, motivated by the Manhattan-world assumption common in indoor/urban scenes. Features from both are trained in a three-stage curriculum and fused for direct equirectangular 3DGS rendering with an RGB-retrieval step for high-frequency detail. Unlike prior panoramic feed-forward methods that depend on multi-view cost volumes or Cartesian triplanes, the cylindrical triplane reduces distortion/aliasing and enables flexible inputs from single to multiple panoramas. Across Matterport3D, Replica, Residential, and 360Loc, CylinderSplat shows state-of-the-art image quality. Ablations show the benefit of the cylindrical triplane. The method is lightweight and fast (≈11.3M params; ~0.29s inference).

**Strengths:**

1. The cylindrical coordinate system choice is well-justified through both theoretical analysis (Manhattan-world assumption) and comprehensive ablations showing clear advantages over Cartesian and spherical alternatives.
2. Extensive experiments are shown across multiple datasets with thorough ablations examining triplane resolution, coordinate systems, initialization strategies, and rendering methods. Consistent improvements are shown in both single-view and multi-view settings, with particularly notable gains in geometric accuracy.
3. Unlike prior cost-volume methods, the attention-based pixel branch handles variable numbers of input views (1 to multiple) without retraining.
4. Good implementation details, clear mathematical formulations, and promise of code release for reproducibility.

**Weaknesses:**

1. Limited novelty: The core contribution is essentially a coordinate system change for existing triplane methods. The dual-branch architecture, attention mechanisms etc. seem to be borrowed from prior work.
2. Manhattan-world assumption limitations: The method seems to be explicitly designed for environments with orthogonal surfaces (indoor/urban scenes). Applicability to natural outdoor scenes, curved structures, or non-Manhattan environments is unclear
3. Training complexity: The three-stage curriculum (pixel branch → volume branch → joint fine-tuning) suggests potential brittleness: Table 4 shows end-to-end training is inferior.

**Questions:**

1. What are some of the scenes/settings where the proposed method is likely to fail?
2. How does performance degrade with larger baselines between input views (>2m)?
3. What is the training time comparison with baselines?

---

> ### Author Response · Authors · 2025-11-22
> **Response to Reviewer 7MCk (Part 1/2)**
>
> We are grateful for the Reviewer 7MCk constructive feedback and for acknowledging the effectiveness of our cylindrical coordinate system and flexible input handling. We are pleased to provide further clarifications and new experiments to address the concerns below.
>
> > ### W1: Limited novelty: The core contribution is essentially a coordinate system change for existing triplane methods. The dual-branch architecture, attention mechanisms, etc., seem to be borrowed from prior work.
>
> **A1:** We thank the reviewer for this comment. Although we build on prior work, our core contribution goes beyond simply changing the coordinate system; it is a systematic redesign of the whole architecture. We showed that the cylindrical triplane effectively matches the geometry of panoramic scenes, solving the distortion issues of other representations.
>
> The key to our performance is that the cylindrical geometry, multi-view strategy, and RGB retrieval are designed to support each other. We modify standard components to handle unique challenges like panoramic geometric distortions and scene completion. As shown in our ablation studies, a simple coordinate swap is not enough; the novelty lies in how we integrated and validated these specific components to make the system work.
>
>
> > ### W2: Applicability to natural outdoor scenes, curved structures, or non-Manhattan environments.
>
> **A2:** We appreciate the reviewer for raising this valid concern. We demonstrate our method's robustness beyond orthogonal structures through two key evidences.
>
> First, we provide additional visualizations of 360Loc's existing curved and outdoor scenes in the supplementary material(Fig.8), confirming effective handling of non-Manhattan geometry.
>
> Second, to further demonstrate robustness in diverse environments, we curated a new large-scale dataset from Google Street View (Kansas City), comprising 8,500 sequences with extreme baselines (20–30m). This dataset covers a wide range of unstructured scenes, including forests and complex urban landscapes. As shown in the supplementary results(Fig.13) and Table R1, our method consistently outperforms baselines in these complex settings, proving its generalization capabilities are not limited to Manhattan-world scenarios.
>
> #### Table R1: Results on Kansas Dataset (20m-30m baseline)
> | Method | PCC $\uparrow$ | WS-PSNR $\uparrow$ | SSIM $\uparrow$ | LPIPS $\downarrow$ |
> | :--- | :---: | :---: | :---: | :---: |
> | OmniScene | 0.55 | 17.08 | 0.44 | 0.35 |
> | Splatter360 | 0.39 | 16.21 | 0.36 | 0.47 |
> | PanSplat | 0.07 | 15.02 | 0.29 | 0.61 |
> | **Ours** | **0.63** | **21.03** | **0.67** | **0.21** |
>
>
> > ### W3 & Q3: Training Complexity and Time Comparison.
>
> **A3:** We clarify that the three-stage training is required only for the initial training on the Matterport3D dataset to ensure robust initialization of the independent Pixel and Volume branches. For all other datasets (e.g., 360Loc, Kansas), we only train the third stage (Joint Training), fine-tuning from the weights trained on Matterport3D.
> As shown in Table R2:
> 1. On Matterport3D: Even with separate stages, the combined per-iteration time of Stage 1 and Stage 2 ($0.81s + 1.13s = 1.94s$) is still faster than a single iteration of PanSplat ($2.17s$). We only need 5 more epochs for the final joint stage.
> 2. On Other Datasets (e.g., 360Loc): Since we only execute the third stage, our training time per iteration (1.98s) is faster than all competing methods (PanSplat 2.17s, Splatter360 2.89s, OmniScene 3.23s).
>
> #### Table R2: Comparison of training efficiency (time per iteration and epochs) against baselines.
> | Method | Training Time (per iteration) | Training Epochs |
> | :--- | :---: | :---: |
> | OmniScene | 3.23s | 10 |
> | Splatter360 | 2.89s | 10 |
> | PanSplat | 2.17s | 10 |
> | **Ours (Stage 1: Pixel Branch)** | **0.81s** | **10** |
> | **Ours (Stage 2: Volume Branch)** | **1.13s** | **10** |
> | **Ours (Stage 3: Joint Training)** | **1.98s** | **5** |
>
> > ### Q1. What are some of the scenes/settings where the proposed method is likely to fail?
>
> **A4:** We appreciate the question. Our method faces challenges in parts of the 360Loc dataset where unavoidable dynamic elements (e.g., the photographer at the nadir, moving pedestrians) are present. Since we do not explicitly optimize for transient objects, these can lead to ghosting artifacts in the synthesized views. We have illustrated these specific cases in the Supplementary Material (Fig.8), noting that while our method shows ghosting, this issue is equally common in competing methods. We will address this in the Limitations section and consider it a direction for future work.

---

> ### Author Response · Authors · 2025-11-22
> **Response to Reviewer 7MCk (Part 2/2)**
>
> > ### Q2. How does performance degrade with larger baselines between input views (>2m)?
>
> **A5:** To address the question regarding performance degradation at larger baselines, we conducted two additional evaluations:
>
> 1. **360Loc with wider baselines (3.0 m, 4.5 m)**.
>
> 2. **Our Kansas Dataset (from W2) with extreme baselines (20 m – 30 m)**
>
> As reported in Tables R1 and R3–R4, our model maintains robust performance even under larger baselines. Crucially, our advantage over competing methods increases substantially as the baseline widens. On the Kansas dataset (20m+ baseline), we outperform the strongest competitor (OmniScene) by +3.95 dB in WS-PSNR. This confirms that CylinderSplat excels in sparse-view synthesis under challenging real-world conditions where other methods degrade rapidly.
>
> #### Table R3: Results on 360Loc (4.5m baseline)
> | Method | PCC $\uparrow$ | WS-PSNR $\uparrow$ | SSIM $\uparrow$ | LPIPS $\downarrow$ |
> | :--- | :---: | :---: | :---: | :---: |
> | OmniScene | 0.84 | 18.12 | 0.53 | 0.41 |
> | Splatter360 | 0.80 | 19.93 | 0.62 | 0.32 |
> | PanSplat | 0.10 | 21.03 | 0.69 | 0.28 |
> | **Ours** | **0.86** | **22.68** | **0.73** | **0.22** |
>
> #### Table R4: Results on 360Loc (3.0m baseline)
> | Method | PCC $\uparrow$ | WS-PSNR $\uparrow$ | SSIM $\uparrow$ | LPIPS $\downarrow$ |
> | :--- | :---: | :---: | :---: | :---: |
> | OmniScene | 0.86 | 21.83 | 0.66 | 0.30 |
> | Splatter360 | 0.82 | 22.65 | 0.71 | 0.24 |
> | PanSplat | 0.11 | 23.47 | 0.75 | 0.22 |
> | **Ours** | **0.86** | **24.50** | **0.79** | **0.18** |

---

### Official Review · Reviewer_5C2X · 2025-11-04

**Soundness:** 3
**Presentation:** 3
**Contribution:** 3
**Rating:** 4
**Confidence:** 5

**Summary:**

The paper proposes CylinderSplat, a feed‑forward 3D Gaussian Splatting framework for 360 panoramas. The system has two branches: (i) a pixel branch that uses attention to aggregate multi‑view features and predict Gaussians for well‑observed pixels, and (ii) a volume branch that introduces per‑camera cylindrical triplanes to complete occlusions and sparsely observed regions. A three‑stage curriculum trains pixel, then volume, next joint fine‑tuning. Color for volume‑branch Gaussians is obtained by RGB retrieval from input panoramas with a simple visibility weighting derived from a depth prior. The paper also implements direct equirectangular 3DGS rasterization without doing cubemap rasterization.

**Strengths:**

- a practical pipeline that avoids heavy cost‑volumes while supporting a variable number of input panoramas via attention.

- Cylindrical triplane is a sensible geometric choice for Manhattan‑style scenes; the paper gives qualitative intuition and ablations showing it outperforms Cartesian and spherical alternatives in this setting.

- The paper measures the effect of each key component: coordinate system, RGB retrieval, multiple per‑camera triplanes, and training curriculum.

- Results are broadly stronger on synthetic datasets, and the method scales from single‑view to multi‑view inputs

**Weaknesses:**

- While the cylindrical triplane is well‑motivated, it can be viewed as a coordinate adaptation of established triplane+3DGS paradigms, rather than a new representation class or learning principle. Much of the performance gain appears to come from engineering side and somehow lacks of technical novelty.

- The transformation from cylindrical local scales to Cartesian scales is kind of approximation without derivation—raising concerns about correctness and whether R and S are consistently transformed.

- PCC is computed against DepthAnywhere (no GT), while UniK3D is used for depth supervision/priors and visibility weighting. Although the models differ, this model‑to‑model comparison weakens claims of geometric accuracy and could bias design choices toward correlating with a particular depth estimator.

- On 360Loc two‑view, gains over a strong panoramic baseline are small (e.g., WS‑PSNR 28.35 vs. 28.14; LPIPS 0.095 vs. 0.127), which tempers the “SOTA” claim outside synthetic domains. This should be carefully justified.

- Competitors like Splatter360 and PanSplat are designed for ≥2 views; duplicating a single view for training creates a non‑native setting that can amplify CylinderSplat’s advantage.

- What exactly is counted in the “inference time” of Table 6 (one forward pass to Gaussians? including rendering of how many targets)?

- Several sampling hyperparameters (e.g.,  N_r, N'_r are left unspecified in the main text; these matter for memory/time.

**Questions:**

- Please provide a derivation (or empirical validation) showing why S' is appropriate for anisotropic Gaussians in 3DGS.

- Can you report geometry vs. ground‑truth depth on any subset (e.g., Replica), or provide alternate metrics less tied to a particular depth estimator?

- How does the method fare on scenes with curved walls or outdoor panoramas where cylindrical alignment is less appropriate?

- Have you tried a learned Gaussian deduplication/merging module (e.g., confidence‑weighted thinning or non‑maximum suppression in splat space) rather than concatenation?

- Since you introduce a direct panoramic rasterizer, can you re‑evaluate OmniScene with the same rasterizer (rather than a cubemap adaptation) to isolate representation vs. rendering pipeline effects?

- Please specify precisely what Table 6 “Inference Time” measures

- Since UniK3D underpins depth supervision and visibility, can you ablate different depth priors (e.g., DepthAnywhere, ZoeDepth) to show robustness?

---

> ### Author Response · Authors · 2025-11-22
> **Response to Reviewer 5C2X (Part 1/4)**
>
> We sincerely thank the Reviewer 5C2X for the detailed feedback and for recognizing our pipeline as "practical" and our cylindrical triplane as a "sensible geometric choice." We address your concerns and questions below.
>
>
> > ### W1: A Simple Adaptation of Established Triplane + 3DGS paradigms?
>
> **A1:** While our approach indeed builds upon the powerful triplane and 3DGS paradigms, we respectfully argue that its core contribution lies in a cohesive architectural redesign that moves beyond a simple coordinate transformation. This is demonstrated through our systematic identification and validation of the cylindrical triplane as the optimal inductive bias for panoramic scenes—a design that fundamentally aligns with the underlying scene geometry to overcome the distortions inherent in Cartesian and the singularities of spherical representations.
> The performance is achieved through a synergistic system where the cylindrical bias, the multi-view strategy, and the rendering pipeline are co-designed. This holistic integration, validated by our ablation studies, is a technical advancement that could not be achieved by a naive coordinate swap in existing frameworks.
>
>
> > ### W2 & Q1: Derivation and Validity of Scale Transformation ($S'$).
>
> **A2:** We thank the reviewer for their rigorous scrutiny. We clarify the design of our coordinate transformation and provide empirical evidence supporting its necessity.
>
> 1. Rotation ($R$).
> First, we clarify that Rotation ($R$) does not undergo coordinate transformation. Consistent with standard 3DGS, our MLP predicts rotation (as quaternions) directly in the global Cartesian coordinate system. The coordinate transformation (Eq. 3 & 4) applies only to the local position offset ($\delta_{local}$) and scale ($S_{local}$).
>
>
> 2. Local Position ($\delta_{local}$) and Scale ($S_{local}$).
> The reviewer is correct that $S'$ involves a local approximation; however, this design strictly follows the standard "volume-bounded" optimization principle used in existing grid-based 3DGS methods. As illustrated in Fig. 2(d) of our paper, methods like OmniScene constrain each Gaussian primitive within a fixed volume element ("Fig. 2(d) Cartesian Volume") defined by the Triplane grid resolution (e.g., dimensions of $2\delta_{x}, 2\delta_{y}, 2\delta_{z}$). The learnable position offsets and the scales are constrained within this grid (e.g., $\pm \delta_{x}$, $\pm \delta_{y}$, $\pm \delta_{z}$). This constraint is critical for ensuring training stability. We apply this same principle to our cylindrical representation. As shown in the "Cylindrical Volume" of Fig. 2(d), our grid units are curvilinear frustums defined by $(2\delta_{r}, 2\delta_{\theta}, 2\delta_{z} )$. Consequently, our network predicts local offsets and scales ($\delta_{local}, S_{local}$) restricted within these local bounds (e.g., $\pm \delta_{r}, \pm \delta_{\theta}, \pm \delta_{z}$). Since the standard 3DGS rasterizer accepts only Cartesian inputs, we must transform these locally predicted parameters into the global Cartesian frame. This necessitates the coordinate transformation for positions (Eq. 3 ) and the Jacobian-based transformation for scales (Eq. 4 ).
>
> Finally, we perform an additional ablation study on Matterport3D (2.0m baseline) comparing our Jacobian-based scaling with predicting Cartesian scales directly within the cylindrical triplane. The results confirm that respecting the local cylindrical geometry through our transformation outperforms the alternative.
>
>
> #### Table R1: Ablation study on scale transformation strategy (Matterport3D, 2.0m baseline).
> | Method | abs_rel $\downarrow$ | rmse $\downarrow$ | delta1 $\uparrow$ | PCC $\uparrow$ | PSNR $\uparrow$ | SSIM $\uparrow$ | LPIPS $\downarrow$ |
> | :--- | :---: | :---: | :---: | :---: | :---: | :---: | :---: |
> | Ours (Cartesian Scale) | 0.22 | 0.55 | 0.73 | 0.82 | 23.54 | 0.82 | 0.18 |
> | **Ours (Cylinder Scale)** | **0.20** | **0.48** | **0.75** | **0.85** | **24.21** | **0.84** | **0.18** |

---

> ### Author Response · Authors · 2025-11-22
> **Response to Reviewer 5C2X (Part 2/4)**
>
> > ### W3 & Q2: Bias in PCC Evaluation.
>
> **A3:** We acknowledge the reviewer's concern regarding the potential bias of using DepthAnywhere as a reference. We originally employed it as a scale-invariant proxy because dense ground truth (GT) depth is unfortunately unavailable or unreliable for the specific test splits of the Replica, Residential, and 360Loc datasets.
>
> To eliminate metric bias, we evaluated against GT depth on Matterport3D, the only dataset which provides GT depth. The results below confirms:
> 1. Consistency: CylinderSplat outperforms all baselines across all standard metrics (AbsRel, RMSE, $\delta_1$) and settings, validating our original PCC conclusions.
> 2. Superiority without Supervision: While competitors (e.g., Splatter360, PanSplat) require GT depth supervision, our method relies solely on RGB images (monocular priors). Outperforming fully supervised baselines without ever seeing GT depth strongly proves that our geometric gains are genuine and robust.
>
> #### Table R2: Quantitative evaluation of geometric accuracy against Ground Truth (GT) depth on the Matterport3D dataset.
> | | **Matterport3D (2.0m)** | | | **Matterport3D (1.5m)** | | | **Matterport3D (1.0m)** | | |
> | :--- | :---: | :---: | :---: | :---: | :---: | :---: | :---: | :---: | :---: |
> | **Method** | **AbsRel** $\downarrow$ | **RMSE** $\downarrow$ | **$\delta_1$** $\uparrow$ | **AbsRel** $\downarrow$ | **RMSE** $\downarrow$ | **$\delta_1$** $\uparrow$ | **AbsRel** $\downarrow$ | **RMSE** $\downarrow$ | **$\delta_1$** $\uparrow$ |
> | OmniScene | 0.36 | 0.77 | 0.37 | 0.34 | 0.72 | 0.44 | 0.17 | 0.38 | 0.79 |
> | Splatter360 | 0.47 | 0.91 | 0.28 | 0.40 | 0.71 | 0.35 | 0.17 | 0.40 | 0.75 |
> | PanSplat | 0.41 | 1.08 | 0.11 | 0.40 | 1.07 | 0.13 | 0.39 | 1.06 | 0.14 |
> | **Ours** | **0.20** | **0.48** | **0.75** | **0.17** | **0.41** | **0.80** | **0.11** | **0.30** | **0.89** |
>
> > ### W4: Performance improvement on 360Loc is slight, tempering “SOTA” claim outside synthetic domains
>
> **A4:** We agree that the improvement on the 360Loc (1.4 m baseline) benchmark is moderate. This is mainly due to:
>
> 1. **Dynamic artifacts**: 360Loc contains unavoidable moving objects (e.g., photographers who hold the camera to take panoramic images, pedestrians in the captured scenes). Our method is not specifically optimized for dynamic content, which restricts potential gains.
>
> 2. **Short-baseline limitation**: Cylinder Triplane primarily benefits geometric completion in occluded regions. At a very short 1.4 m baseline, view overlap is high, so the advantage of our geometry hallucination is inherently limited.
>
> To more fully support the SOTA claim in real-world settings, we conducted two additional evaluations focusing on challenging wide-baseline scenarios where geometric completion matters most:
>
> 1. **360Loc with wider baselines (3.0 m, 4.5 m)**.
>
> 2. **A new large-scale real-world dataset from Google Street View (Kansas City)**, with 8,500 sequences and extreme baselines (20–30 m) but fewer dynamic objects.
>
> As reported in Tables R3–R5, although the gains at 1.4 m are small, our advantage increases substantially with wider baselines. On the Kansas dataset (20 m+ baseline), we outperform the strongest competitor (OmniScene) by +3.95 dB WS-PSNR, confirming that CylinderSplat excels in sparse-view synthesis under challenging real-world conditions.
>
> #### Table R3: Results on Kansas Dataset (20m-30m baseline).
> | Method | PCC $\uparrow$ | WS-PSNR $\uparrow$ | SSIM $\uparrow$ | LPIPS $\downarrow$ |
> | :--- | :---: | :---: | :---: | :---: |
> | OmniScene | 0.55 | 17.08 | 0.44 | 0.35 |
> | Splatter360 | 0.39 | 16.21 | 0.36 | 0.47 |
> | PanSplat | 0.07 | 15.02 | 0.29 | 0.61 |
> | **Ours** | **0.63** | **21.03** | **0.67** | **0.21** |
>
> #### Table R4: Results on 360Loc(4.5m baseline) with wider baselines.
> | Method | PCC $\uparrow$ | WS-PSNR $\uparrow$ | SSIM $\uparrow$ | LPIPS $\downarrow$ |
> | :--- | :---: | :---: | :---: | :---: |
> | OmniScene | 0.84 | 18.12 | 0.53 | 0.41 |
> | Splatter360 | 0.80 | 19.93 | 0.62 | 0.32 |
> | PanSplat | 0.10 | 21.03 | 0.69 | 0.28 |
> | **Ours** | **0.86** | **22.68** | **0.73** | **0.22** |
>
> #### Table R5: Results on 360Loc(3.0m baseline) with wider baselines.
> | Method | PCC $\uparrow$ | WS-PSNR $\uparrow$ | SSIM $\uparrow$ | LPIPS $\downarrow$ |
> | :--- | :---: | :---: | :---: | :---: |
> | OmniScene | 0.86 | 21.83 | 0.66 | 0.30 |
> | Splatter360 | 0.82 | 22.65 | 0.71 | 0.24 |
> | PanSplat | 0.11 | 23.47 | 0.75 | 0.22 |
> | **Ours** | **0.86** | **24.50** | **0.79** | **0.18** |

---

> ### Author Response · Authors · 2025-11-22
> **Response to Reviewer 5C2X (Part 3/4)**
>
> > ### W5: Comparison with Splatter360 and PanSplat in the single-view setting is unfair
>
> **A5**: We appreciate the reviewer’s observation and agree that Splatter360 and PanSplat are designed for ≥2 input views. Our intention in including them is to highlight architectural flexibility across panoramic 3DGS-based methods.
>
> Splatter360 and PanSplat are the closest panoramic 3DGS baselines, and although they are built for stereo or multi-view inputs, evaluating them under the same single-view condition exposes an important distinction: cost-volume–based designs inherently depend on multi-view geometry, whereas our attention-driven architecture naturally supports single-view (N=1) inputs without modification. This comparison demonstrates that CylinderSplat offers greater input flexibility while still delivering strong reconstruction performance.
>
> To ensure fairness, we also report comparisons with OmniScene, a method explicitly designed for multi-view, surrounding-input settings. Our clear performance improvement (e.g., +1.2 dB on Matterport3D at a 2.0 m baseline) further validates the effectiveness of our cylindrical representation under its native operating regime.
>
> > ### W6. What exactly is counted in the “inference time” of Table 6?
>
> **A6**: In Table 6, "Inference Time" measures the end-to-end latency for a single forward pass that outputs Gaussians and renders them into one panorama when batch size = 2.
>
>
> > ### W7. Undefined hyperparameters (e.g., N_r, N'_r), how do they affect memory/time.
>
> **A7**: We appreciate the reviewer for highlighting the omission of these specific hyperparameters. We have now specified the sampling resolutions corresponding to the $(r, z, \theta)$ axes for both the Cross-Plane Attention sampling ($N_r$) and the Triplane-to-Image Attention sampling ($N'_r$).To analyze their impact on efficiency, we conducted an ablation study as shown in Table R6: we fixed $N'_r$ at $(8, 32, 64)$ when varying $N_r$, and fixed $N_r$ at $(16, 64, 128)$ when varying $N'_r$. As shown in Table R6, while increasing sampling density consistently improves quality, it incurs a cost in GPU memory and inference latency. Consequently, we selected the configuration of $N_r=(16, 64, 128)$ and $N'_r=(8, 32, 64)$ (highlighted in bold) as it strikes the optimal balance between performance and computational efficiency.
>
> #### Table R6: Ablation study on sampling hyperparameters ($N_r$ and $N'_r$) on Matterport3D 2.0m baseline.
>
> | Configuration $(r, \theta, z)$ | PCC $\uparrow$ | WS-PSNR $\uparrow$ | SSIM $\uparrow$ | LPIPS $\downarrow$ | Memory | Time |
> | :--- | :---: | :---: | :---: | :---: | :---: | :---: |
> | $N_r (16 \times 32 \times 64)$ | 0.72 | 21.18 | 0.71 | 0.24 | 15GB | 0.26s |
> | **$N_r (16 \times 64 \times 128)$** | **0.78** | **22.17** | **0.78** | **0.21** | **17GB** | **0.29s** |
> | $N_r (16 \times 128 \times 256)$ | 0.80 | 23.88 | 0.82 | 0.19 | 25GB | 0.34s |
> | $N'_r (8 \times 16 \times 32)$ | 0.77 | 22.08 | 0.73 | 0.28 | 16GB | 0.27s |
> | **$N'_r (8 \times 32 \times 64)$** | **0.78** | **22.17** | **0.78** | **0.21** | **17GB** | **0.29s** |
> | $N'_r (8 \times 64 \times 128)$ | 0.79 | 22.46 | 0.80 | 0.21 | 18GB | 0.31s |
>
> > ### Q3: Performance on Scenes with Curved Walls or Outdoor Panoramas.
>
> **A8**: To address concerns regarding non-Manhattan geometries, we have included additional visualizations in the supplementary material, specifically targeting outdoor scenes from the 360Loc dataset(Fig.8) and complex urban/forest environments from the Kansas dataset(Fig.13). These qualitative results demonstrate that even in scenarios featuring curved structures or unstructured outdoor geometry, our method maintains robust performance comparable to state-of-the-art baselines.

---

> ### Author Response · Authors · 2025-11-22
> **Response to Reviewer 5C2X (Part 4/4)**
>
> > ### Q4. Have you tried a learned Gaussian deduplication/merging module (e.g., confidence‑weighted thinning or non‑maximum suppression in splat space) rather than concatenation?
>
> **A9**: We sincerely appreciate this insightful suggestion. To explore this direction, we implemented and evaluated two specific strategies: Confidence-Weighted Thinning and Depth-Guided Non-Maximum Suppression (NMS).
>
> As shown in Table R7, our findings are as follows:
>
> 1. **Confidence‑Weighted Thinning**: We attempted to merge tightly packed Gaussians by learning confidence weights via a softmax mechanism. However, we found that the network struggled to learn reliable confidence scores for fusion, leading to visual artifacts (distortions and holes) and a performance drop compared to our original concatenation baseline.
> 2. **Depth-Guided NMS**: Conversely, this strategy proved effective. By enhancing the opacity of Gaussians located near the network-predicted depth, suppressing those with significant depth deviation, and filtering out low-confidence primitives, we achieved improvements in both geometric accuracy and rendering quality. We would like to incorporate this in our final version.
>
> #### Table R7: Comparison of different Gaussian fusion strategies on Matterport3D 2.0m baseline.
> | Method | PCC $\uparrow$ | WS-PSNR $\uparrow$ | SSIM $\uparrow$ | LPIPS $\downarrow$ |
> | :--- | :---: | :---: | :---: | :---: |
> | Confidence‑Weighted Thinning | 0.84 | 22.52 | 0.75 | 0.24 |
> | **Non‑Maximum Suppression** | **0.87** | **24.64** | **0.84** | **0.15** |
> | Ours (Original) | 0.85 | 24.21 | 0.83 | 0.17 |
>
>
> > ### Q5: Re-evaluating OmniScene with Direct Panoramic Rasterizer.
>
> **A10**: To isolate the impact of the rendering pipeline from the geometric representation, we re-evaluate OmniScene using our direct panoramic rasterizer. As shown in Table R8, replacing the cubemap renderer with our direct rasterizer yields comparable overall performance. This mirrors the conclusion in our main paper (Table 9), where we demonstrated that switching CylinderSplat to a cubemap renderer had negligible impact on quality. Thus, we conclude that the direct rasterizer primarily contributes to inference speed, while the superior reconstruction quality is driven by our Cylindrical Triplane representation.
>
> #### Table R8: Comparison of OmniScene performance using Cubemap vs. Direct Panoramic Rasterizer on Matterport3D 2.0m baseline.
> | Method | AbsRel $\downarrow$ | RMSE $\downarrow$ | $\delta_1$ $\uparrow$ | PCC $\uparrow$ | PSNR $\uparrow$ | SSIM $\uparrow$ | LPIPS $\downarrow$ |
> | :--- | :---: | :---: | :---: | :---: | :---: | :---: | :---: |
> | OmniScene (Cubemap) | 0.36 | 0.77 | 0.37 | 0.73 | 22.75 | 0.71 | 0.24 |
> | OmniScene (Direct Raster) | 0.29 | 0.69 | 0.41 | 0.76 | 22.19 | 0.76 | 0.24 |
>
> > ### Q6. Ablate different depth priors (e.g., DepthAnywhere, ZoeDepth) to show robustness.
>
> **A11**: We appreciate the reviewer's suggestion to evaluate the robustness of our method against different depth priors. To demonstrate the robustness of our framework, we substituted our default UniK3D prior with DepthAnywhere and UniFuse. We specifically included UniFuse as it serves as the depth prior for both PanSplat and Splatter360.
>
> Table R9 demonstrates that our method maintains consistent performance advantages across different depth priors. Even when utilizing the exact same UniFuse prior as PanSplat and Splatter360, our approach yields superior quality. Crucially, we achieve these results without ground truth (GT) depth supervision; in contrast, both Splatter360 and PanSplat rely on UniFuse for priors and additionally require GT depth as supervision during training. We did not include ZoeDepth in our experiments because it is explicitly designed for pinhole cameras, rendering it less suitable than the panoramic-aware models we selected.
>
> #### Table R9: Different depth prior on Matterport3D 2.0m baseline.
>
> | Method | AbsRel $\downarrow$ | RMSE $\downarrow$ | $\delta_1$ $\uparrow$ | PCC $\uparrow$ | PSNR $\uparrow$ | SSIM $\uparrow$ | LPIPS $\downarrow$ |
> | :--- | :---: | :---: | :---: | :---: | :---: | :---: | :---: |
> | OmniScene | 0.36 | 0.77 | 0.37 | 0.73 | 22.75 | 0.71 | 0.24 |
> | Splatter360 | 0.47 | 0.91 | 0.28 | 0.68 | 21.31 | 0.74 | 0.28 |
> | PanSplat | 0.41 | 1.08 | 0.11 | 0.71 | 20.56 | 0.77 | 0.26 |
> | **Ours (DepthAnywhere)** | **0.34** | **0.66** | **0.60** | **0.78** | **23.08** | **0.78** | **0.22** |
> | **Ours (UniFuse)** | **0.31** | **0.64** | **0.62** | **0.78** | **23.31** | **0.81** | **0.21** |
> | **Ours (UniK3D)** | **0.20** | **0.48** | **0.75** | **0.85** | **24.21** | **0.83** | **0.17** |

---

> > ### Comment · Reviewer_5C2X · 2025-11-22
> >
> > I would like to thank the authors for their efforts in addressing the raised concerns. Those have addressed most of my concerns and I encourge the authors to update those results to the paper, in order to meet the high standard of the ICLR. I have raised my rating.

---

> > > ### Author Response · Authors · 2025-11-23
> > >
> > > Dear Reviewer 5C2X,
> > >
> > > We sincerely thank you for your time and effort in re-evaluating our work. We’re excited to receive your positive feedback and truly appreciate you raising our score!
> > >
> > > Your constructive comments—especially regarding the validation of our scale transformation, a learned Gaussian deduplication/merging module, the fairness of comparisons against ground truth depth, and the clarification of our architecture—have been crucial in enhancing the rigor and quality of this paper. We appreciate your thorough review, which motivated us to include more comprehensive evidence and clearer explanations.
> > >
> > > All new findings from this rebuttal have been added to the supplementary material. Meanwhile, we are currently refining the manuscript to seamlessly incorporate these updates and aim to deliver the most polished and complete version of our work in the final paper.
> > >
> > > Thank you once again for your valuable contribution to our research!
> > >
> > > Best regards,
> > >
> > > The Authors

---

### Official Review · Reviewer_K1rt · 2025-11-11

**Soundness:** 3
**Presentation:** 3
**Contribution:** 3
**Rating:** 6
**Confidence:** 3

**Summary:**

The authors propose a panoramic novel view synthesis method using 3D Gaussian Splatting (3DGS). They introduce a cylindrical triplane representation combined with a dual-branch design: an attention-based, cost-volume-free pixel branch that parameterizes Gaussians from the inputs, and a per-camera local cylindrical-triplane volume branch that handles occlusions. On both synthetic and real datasets, the method achieves competitive image quality (WS-PSNR, SSIM, LPIPS) and geometry metrics (PCC) with strong efficiency.

**Strengths:**

1. Originality. Introduces a cylindrical triplane representation specifically designed for panoramic 3D Gaussian Splatting. Proposes a three-stage curriculum training strategy (pixel, volume, joint) that significantly enhances reconstruction in occluded and sparsely viewed regions.

2. Quality. Provides comprehensive evaluations across multiple panoramic and indoor datasets with consistent improvements over prior methods. Includes thorough ablation studies demonstrating the effectiveness of the cylindrical triplane and each component.

3. The paper is well-organized and easy to follow.

4. Significance. Effectively addresses the blurriness and occlusion artifacts found in previous sparse-view panoramic methods.

**Weaknesses:**

1. The method relies heavily on a strong depth prior, but the paper does not analyze its robustness or potential failure cases.
2. The domain bias is significant—experiments focus primarily on indoor, Manhattan-world panoramas. Analysis of non-Manhattan or outdoor environments would strengthen the generality claims.
3. The paper does not discuss or address artifacts near seams and poles, which are common in panoramic representations and could impact real-world performance.

**Questions:**

Using a foundation model–based depth prior as a reference for geometry evaluation raises concerns about circular validation and fairness in comparisons.

---

> ### Author Response · Authors · 2025-11-22
> **Response to Reviewer K1rt (Part 1/2)**
>
> We sincerely thank the Reviewer K1rt for the positive assessment and for recognizing the originality and significance of our work in addressing panoramic occlusions. We appreciate the constructive feedback regarding depth robustness, domain bias and artifacts, which we address below with new experimental evidence.
>
> > ### Q1: Using a foundation model–based depth prior as a reference for geometry evaluation raises concerns about circular validation and fairness in comparisons.
>
> **A1:** We acknowledge the reviewer's concern regarding the potential bias of using DepthAnywhere as a reference. We originally employed it as a scale-invariant proxy because dense ground truth (GT) depth is unfortunately unavailable or unreliable for the specific test splits of the Replica, Residential, and 360Loc datasets.
>
> To eliminate metric bias, we evaluate against GT depth on Matterport3D, the only dataset which provides GT depth. The results below confirms:
> 1. Consistency: CylinderSplat outperforms all baselines across all standard metrics (AbsRel, RMSE, $\delta_1$) and settings, validating our original PCC conclusions.
> 2. Superiority without Supervision: While competitors (e.g., Splatter360, PanSplat) require GT depth supervision, our method relies solely on RGB images (monocular priors). Outperforming fully supervised baselines without ever seeing GT depth strongly proves that our geometric gains are genuine and robust.
>
> #### Table R1: Quantitative evaluation of geometric accuracy against Ground Truth (GT) depth on the Matterport3D dataset.
> | | **Matterport3D (2.0m)** | | | **Matterport3D (1.5m)** | | | **Matterport3D (1.0m)** | | |
> | :--- | :---: | :---: | :---: | :---: | :---: | :---: | :---: | :---: | :---: |
> | **Method** | **AbsRel** $\downarrow$ | **RMSE** $\downarrow$ | **$\delta_1$** $\uparrow$ | **AbsRel** $\downarrow$ | **RMSE** $\downarrow$ | **$\delta_1$** $\uparrow$ | **AbsRel** $\downarrow$ | **RMSE** $\downarrow$ | **$\delta_1$** $\uparrow$ |
> | OmniScene | 0.36 | 0.77 | 0.37 | 0.34 | 0.72 | 0.44 | 0.17 | 0.38 | 0.79 |
> | Splatter360 | 0.47 | 0.91 | 0.28 | 0.40 | 0.71 | 0.35 | 0.17 | 0.40 | 0.75 |
> | PanSplat | 0.41 | 1.08 | 0.11 | 0.40 | 1.07 | 0.13 | 0.39 | 1.06 | 0.14 |
> | **Ours** | **0.20** | **0.48** | **0.75** | **0.17** | **0.41** | **0.80** | **0.11** | **0.30** | **0.89** |
>
>
>
> > ### W1: The method relies heavily on a strong depth prior, but the paper does not analyze its robustness or potential failure cases.
>
> **A2:** We appreciate the reviewer's suggestion to evaluate the robustness of our method against different depth priors. To demonstrate the robustness of our framework, we substitute our default UniK3D prior with DepthAnywhere and UniFuse, and provided a more comprehensive depth evaluation(AbsRel, RMSE and $\delta_1$). We specifically included UniFuse as it serves as the depth prior for both PanSplat and Splatter360.
>
> Table R2 demonstrates that our method maintains consistent performance advantages across different depth priors. Even when utilizing the exact same UniFuse prior as PanSplat and Splatter360, our approach yields superior quality. Crucially, we achieve these results without ground truth (GT) depth supervision; in contrast, both Splatter360 and PanSplat rely on UniFuse for priors and additionally require GT depth as supervision during training.
>
> #### Table R2: Performance with Different Depth Priors (Matterport3D, 2.0m baseline).
>
> | Method | AbsRel $\downarrow$ | RMSE $\downarrow$ | $\delta_1$ $\uparrow$ | PCC $\uparrow$ | PSNR $\uparrow$ | SSIM $\uparrow$ | LPIPS $\downarrow$ |
> | :--- | :---: | :---: | :---: | :---: | :---: | :---: | :---: |
> | OmniScene | 0.36 | 0.77 | 0.37 | 0.73 | 22.75 | 0.70 | 0.24 |
> | Splatter360 | 0.47 | 0.91 | 0.28 | 0.68 | 21.31 | 0.74 | 0.28 |
> | PanSplat | 0.41 | 1.08 | 0.11 | 0.71 | 20.56 | 0.77 | 0.26 |
> | **Ours (DepthAnywhere)** | **0.34** | **0.66** | **0.60** | **0.78** | **23.08** | **0.78** | **0.22** |
> | **Ours (UniFuse)** | **0.31** | **0.64** | **0.62** | **0.78** | **23.31** | **0.81** | **0.21** |
> | **Ours (UniK3D)** | **0.20** | **0.48** | **0.75** | **0.85** | **24.21** | **0.83** | **0.17** |

---

> ### Author Response · Authors · 2025-11-22
> **Response to Reviewer K1rt (Part 2/2)**
>
> > ### W2: The domain bias is significant—experiments focus primarily on indoor, Manhattan-world panoramas. Analysis of non-Manhattan or outdoor environments would strengthen the generality claims.
>
> **A3:** We appreciate the reviewer for raising this valid concern. We demonstrate our method's robustness beyond indoor, Manhattan-world panoramas through two key pieces of evidence.
>
> First, we provide additional visualizations of 360Loc's existing curved and outdoor scenes in the supplementary material(Fig.8), confirming effective handling of non-Manhattan geometry.
>
> Second, to further demonstrate robustness in diverse environments, we curated a new large-scale dataset from Google Street View (Kansas City), comprising 8,500 sequences with extreme baselines (20–30m). This dataset covers a wide range of unstructured scenes, including forests and complex urban landscapes. As shown in the supplementary results(Fig.13) and Table R3, our method consistently outperforms baselines in these complex settings, proving its generalization capabilities are not limited to Manhattan-world scenarios.
>
> #### Table R3: Results on Kansas Dataset (20m-30m baseline).
> | Method | PCC $\uparrow$ | WS-PSNR $\uparrow$ | SSIM $\uparrow$ | LPIPS $\downarrow$ |
> | :--- | :---: | :---: | :---: | :---: |
> | OmniScene | 0.55 | 17.08 | 0.44 | 0.35 |
> | Splatter360 | 0.39 | 16.21 | 0.36 | 0.47 |
> | PanSplat | 0.07 | 15.02 | 0.29 | 0.61 |
> | **Ours** | **0.63** | **21.03** | **0.67** | **0.21** |
>
> > ### W3. The paper does not discuss or address artifacts near seams and poles, which are common in panoramic representations and could impact real-world performance.
>
> **A4:** We thank the reviewer for their rigorous scrutiny. We address the concerns regarding artifacts near seams and poles as follows:
>
> 1. **Seams**: The Cylindrical Triplane is logically circular in the $\theta$ dimension; queries at $\theta=0$ and $\theta=2\pi$ access the exact same physical location in the coordinate system, rendering our method naturally seamless. To quantify this, we evaluate the Left-Right Consistency Error (LRCE) (the mean pixel difference between the left and right boundaries). As shown in Table R4, our method achieves an error that is an order of magnitude lower than competing methods (e.g., 0.025 vs. PanSplat's 0.088), confirming superior continuity.
>
> 2. **Poles**: As visualized in Fig.9 of the supplementary material, our Cylindrical Triplane leverages a geometric prior that optimally aligns with real-world depth distributions: it prioritizes sampling at the poles for regions with small radii (close to the camera) while focusing on the central horizon for regions with large radii (far from the camera). Furthermore, by maintaining uniform resolution along the $Z$-axis, our representation ensures consistent detail for the zenith and nadir regions, allowing the geometric representation itself to handle polar areas robustly, as shown in (Fig. 4, Fig. 5, Fig. 6, and Fig. 8).
>
>
> #### Table R4: Left-Right Consistency Error (LRCE) $\downarrow$.
> | Method | Matterport3D(2.0m) | Matterport3D(1.5m) | Matterport3D(1.0m) |
> | :--- | :---: | :---: | :---: |
> | OmniScene | 0.291 | 0.325 | 0.299 |
> | Splatter360 | 0.165 | 0.146 | 0.128 |
> | PanSplat | 0.137 | 0.119 | 0.088 |
> | **Ours** | **0.025** | **0.024** | **0.023** |

---

### Meta-Review · Area_Chair_bti2 · 2026-01-03

**Summary:**

Most reviewers admit that the pipeline proposed is reasonable and achieves good results on image quality. However, concerns about the limitation on reconstucting only indoor and orthogonal surfaces are raised by more than one reviewers. Also, some reviewers ask for more clarfication about the influence of depth prior. Most concerns have been attentively responsed by the authors. The overall quality of this paper is above the borderline of acceptance.

**Reviewer Concerns:**

Most concerns are well responded.

**Reviewer Scores:**

Most reviewers are not likely to change their scores. The reviewer 5C2X have shown great responsibility and  professionalism even though he is the only reviewer giving negative rating. The authors have responded to every concern; however, the reviewer clearly has more refined requirements for this work.

---

### Decision · Program_Chairs · 2026-01-26

Accept (Poster)